# Cross Aggregation Transformer for Image Restoration

**Zheng Chen**[1], **Yulun Zhang**[2], **Jinjin Gu**[3,4], **Yongbing Zhang**[5], **Linghe Kong**[1]*, **Xin Yuan**[6]

[1]Shanghai Jiao Tong University, [2]ETH Zürich, [3]Shanghai AI Laboratory,
[4]The University of Sydney, [5]Harbin Institute of Technology (Shenzhen), [6]Westlake University

## Abstract

Recently, Transformer architecture has been introduced into image restoration to replace convolution neural network (CNN) with surprising results. Considering the high computational complexity of Transformer with global attention, some methods use the local square window to limit the scope of self-attention. However, these methods lack direct interaction among different windows, which limits the establishment of long-range dependencies. To address the above issue, we propose a new image restoration model, Cross Aggregation Transformer (CAT). The core of our CAT is the Rectangle-Window Self-Attention (Rwin-SA), which utilizes horizontal and vertical rectangle window attention in different heads parallelly to expand the attention area and aggregate the features cross different windows. We also introduce the Axial-Shift operation for different window interactions. Furthermore, we propose the Locality Complementary Module to complement the self-attention mechanism, which incorporates the inductive bias of CNN (*e.g.*, translation invariance and locality) into Transformer, enabling global-local coupling. Extensive experiments demonstrate that our CAT outperforms recent state-of-the-art methods on several image restoration applications. The code and models are available at <https://github.com/zhengchen1999/CAT>.

## 1 Introduction

Image restoration is a long-term low-level vision problem, aimed at recovering high-quality (HQ) images from low-quality (LQ) counterparts. Depending on the type of degradation, there are many sub-problems, such as super-resolution (SR), image denoising, and JPEG compression artifact reduction. In recent decades, many researchers have proposed various theories and methods to solve this problem, *e.g.*, inverse filtering and wiener filtering. Meanwhile, with the increase of computing resources, neural networks have become the mainstream of image restoration, especially convolutional neural network (CNN) [10, 11, 50], which achieve impressive performance over traditional methods.

Recently, some researchers have been using Transformer proposed in natural language processing (NLP) to replace CNN and achieving excellent performance in multiple vision tasks, such as image classification [13], semantic segmentation [12], and object detection [41]. The core component of Transformer is the multi-head self-attention (MHSA), which can effectively model long-range dependencies, while CNN is more inclined to extract local features. Although Transformer can effectively handle dependencies between tokens, its complexity is quadratic with the image resolution ($H \times W$), *i.e.*, $\mathcal{O}((HW)^2)$, which limits its application to high-resolution image tasks (including image restoration). To apply Transformer in image restoration, some methods [5, 21, 44] propose to divide the image into smaller independent patches (*e.g.*, $8 \times 8$) and perform attention separately or apply self-attention (SA) in the feature dimension instead of the spatial dimension, thus achieving linear complexity. However, the square window lacks inter-window interaction, leading to the slow

---

*Corresponding author: Linghe Kong, linghe.kong@sjtu.edu.cn

36th Conference on Neural Information Processing Systems (NeurIPS 2022).

increase of the receptive field. Moreover, the channel-wise attention mechanism may lose some spatial information. All these problems restrict the performance of the Transformer in image restoration.

To solve the above problems of Transformer application in image restoration tasks, we propose a new image restoration Transformer model named Cross Aggregation Transformer (CAT). Our CAT utilizes local window self-attention to restrict the computational complexity and aggregates features cross different windows to expand the receptive field. The key part of our CAT is a new self-attention mechanism, named **Rectangle-Window Self-Attention** (Rwin-SA). Specifically, the Rwin-SA performs the attention operation in a separate rectangle window ($sh \neq sw$) rather than the square ($sh=sw$) one, where $sh$ and $sw$ are window height and width. And we divide multiple heads into two groups and perform horizontal ($sh<sw$) and vertical ($sh>sw$) rectangle window self-attention in parallel. This method effectively expands the attention area and aggregates different features in horizontal and vertical directions while keeping the computational complexity unchanged. Moreover, inspired by CSwin [12], which performs cross-shaped window self-attention, we fix the length of one side of the rectangle window to be $H$ or $W$ (image resolution). Then the window becomes a stripe along the axis, called axial rectangle window. It can generate more window interactions and obtain a larger attention area than the regular rectangle window.

We also propose a new shift operation named **Axial-Shift Operation**, which is applied between consecutive Rwin-SA, to further increase the interaction of the different windows. Like our Rwin-SA, axial-shift is also divided into horizontal and vertical directions, which acts on the corresponding window respectively. Compared with the shift operation in Swin Transformer [23], our axial-shift explicitly realizes the interaction between horizontal-horizontal or vertical-vertical windows, and implicitly enables the interaction between horizontal and vertical windows.

Furthermore, we propose **Locality Complementary Module** (LCM), a convolution operation that it performed on the value ($V$) in the self-attention mechanism in parallel with the attention part. Compared with Transformer that mainly builds global long-range dependencies, CNN has the characteristics of translation invariance and locality, and can effectively capture 2D local structures of the image (*e.g.*, corners and edges). The LCM can complements Rwin-SA with locality information, enabling the coupling of global (self-attention) and local (convolution) information.

Based on the above three designs, our CAT realizes the feature aggregation cross different windows, thus possessing long-range dependencies modeling capability. We apply our CAT to several classic image restoration tasks: image SR, JPEG compression artifacts reduction, and real image denoising. Extensive experiments demonstrate the superiority of our method. Our contributions are as follows:

- We propose a new image restoration Transformer model named Cross Aggregation Transformer (CAT). The CAT utilizes the window self-attention and aggregates the features cross different windows, thus achieving a large receptive field with linear complexity.

- We propose a novel self-attention mechanism, named Rwin-SA, using rectangle window self-attention with axial-shift operation. Moreover, we propose the locality complementary module to realize the coupling of global and local information.

- We apply our CAT to classic image restoration tasks: image SR, JPEG compression artifact reduction, and real image denoising. Extensive experiments show that our proposed model achieves state-of-the-art performance both quantitatively and visually.

## 2 Related Work

**Image Restoration.** CNNs have been achieving impressive performance over the traditional approaches and becoming the mainstream method in the field of image restoration since Dong *et al.* proposed SRCNN [11] and ARCNN [10] for image SR and JPEG compression artifact reduction, respectively. Currently, many efforts have been devoted to building very deep CNNs to improve performance through multiple methods, *e.g.*, residual learning strategies [15], dense connection [16], dropout [19], and UNet architecture [34]. Meanwhile, spatial and channel attention mechanisms [51, 31, 52, 20, 6] are utilized to make the network more focused on specific information in the feature space. However, most CNN-based methods are still hard to capture global dependencies.

**Vision Transformer.** Vision Transformer (ViT) achieves surprising results in high-level vision tasks [13, 41]. Meanwhile, many efficient attention mechanisms have been proposed to improve the performance of ViT. Swin Transformer [23] uses local window attention and realizes the interaction between windows through shift operations. CSwin [12] proposes cross-shaped window attention, and

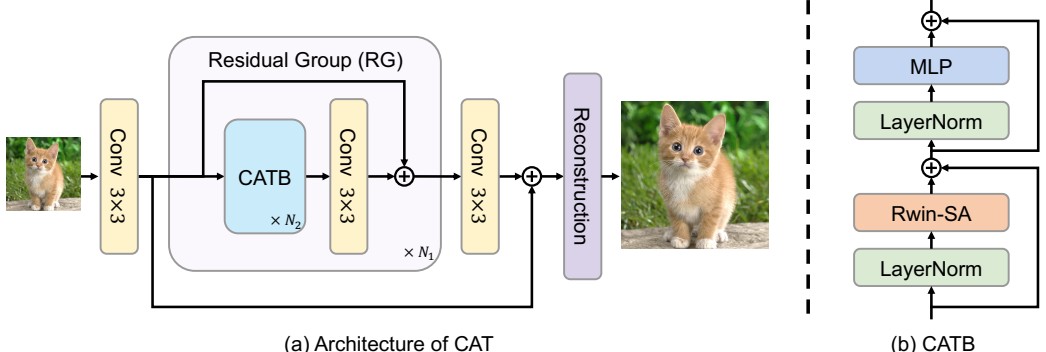

Figure 1: (a) The architecture of our CAT. (b) Illustration of CATB.

Twins [7] adopt a combination of global and local attention. Inspired by the success of Transformer in high-level tasks, some works have been trying to apply Transformer to low-level vision tasks, such as IPT [5]. However, employing original ViT [13] for restoration tasks is hard since the complexity of full self-attention is quadratic to image size. Some methods have been proposed to solve this problem. SwinIR [21], based on Swin Transformer, utilizes an $8{\times}8$ window to divide the image and executes self-attention in each window separately. It also uses the shift operation. Uformer [39] also applies $8{\times}8$ local window and introduces UNet architecture to capture much more global dependencies. Restormer [44] calculates cross-covariance across channel dimensions rather than spatial ones, making the complexity of SA linear with image resolution. However, the $8{\times}8$ square window limits the receptive field of the Transformer even with the shift operation. And the channel-wise attention mechanisms may lose spatial information. Furthermore, above methods do not effectively combine convolution with attention mechanisms, thus limiting model capabilities.

## 3 Method

To build an efficient Transformer model suitable for image restoration, we propose a rectangle-window self-attention (Rwin-SA) to replace vanilla multi-head self-attention mechanism in ViT [13] and form the cross aggregation Transformer block (CATB). Then we use RCAN [51] as the backbone and replace the residual channel attention block (RCAB) proposed in RCAN with our CATB to build the model cross aggregation Transformer (CAT), which is illustrated in Fig. 1. We first describe the overall structure of CAT and then introduce the three core designs of the CATB, including rectangle-window self-attention, axial-shift operation, and locality complementary module.

### 3.1 CAT Architecture

As shown in Fig. 1, following RCAN [51], our proposed CAT consists of three modules: shallow feature extraction, deep feature extraction, and reconstruction part. For a low-quality input image $I_{LQ}{\in}\mathbb{R}^{H\times W\times C_{in}}$, we leverage one convolution layer as shallow feature extraction to get the low-level feature $F_0{\in}\mathbb{R}^{H\times W\times C}$, where $H, W, C_{in}, C$ are the image height, width, input channel, and the feature number, respectively. Then, the shallow feature is processed by the deep feature extraction module, which is composed of $N_1$ residual groups (RG) [51] and one convolution layer, and extracted the deep feature (denoted as $F_{DF}{\in}\mathbb{R}^{H\times W\times C}$). The convolution layer here is employed to aggregate previously extracted features from RG. Moreover, a residual connection is applied in the deep feature extraction part to make the training stable. We use CATB to replace RCAB [51] in RG, as we mentioned above. So, RG consists of $N_2$ cross aggregation Transformer blocks and a convolution layer that can introduce the unique properties (translation invariance and locality) of CNN into the output of Transformer blocks. In addition, a residual strategy is also applied in each RG.

Finally, we can get the high-quality output image $I_{HQ}{\in}\mathbb{R}^{H\times W\times C_{out}}$ through the reconstruction part, where $C_{out}$ is the output dimension. And following SwinIR [21], the composition of reconstruction module is different for different image restoration tasks. More specifically, for image SR, a sub-pixel convolution layer [36] is used to upsample the deep feature $F_{DF}$ to the same size of the high-resolution output and there is a convolution layer before and after the upsampling module to aggregate the features. For JPEG compression artifact reduction, the reconstruction is just a convolution layer to adjust the channel dimension of the $F_{DF}$ from $C$ to $C_{out}$. The low-quality input $I_{LQ}$ is then added to the convolution output to produce a high-quality output $I_{HQ}$. This residual learning can accelerate network model convergence. Moreover, for real image denoising, following Restormer [44], we apply our proposed CATB to the UNet architecture [34] to construct CAT.

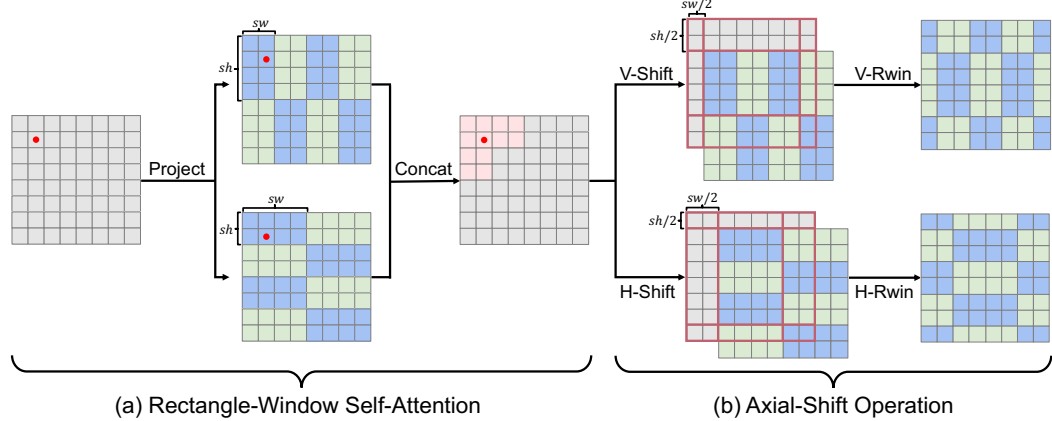

Figure 2: (a) Illustration of rectangle-window self-attention mechanism. The $sh$ and $sw$ are the window height and width for H-Rwin and V-Rwin. For a pixel (red dot), the attention area is the pink region of the middle feature map. (b) Illustration of axial-shift operation.

## 3.2 Cross Aggregation Transformer Block

The cross aggregation Transformer block (CATB) is the key part of our CAT, shown in Fig. 1, which is more suitable for image restoration tasks than the previous Transformer structures [21, 44]. It utilizes a new attention mechanism with three novel designs: rectangle-window self-attention, axial-shift operation, and locality complementary module. We will describe these parts in detail below.

**Rectangle-Window Self-Attention**. To alleviate the drawbacks of previous methods [5, 21, 44], we propose a new window attention mechanism as shown in Fig. 2(a), called rectangle-window self-attention (Rwin-SA). Rwin-SA uses the rectangle window ($sh \neq sw$) rather than the square ($sh = sw$) one, where $sh$ and $sw$ represent the height and width of the rectangle. Moreover, we divide the rectangle window into the horizontal one ($sh < sw$, denoted as H-Rwin) and the vertical one ($sh > sw$, denoted as V-Rwin), and employ them to different attention heads in parallel.

Specifically, given the input $X \in \mathbb{R}^{H \times W \times C}$, we execute the attention operation $M$ (the heads number) times in parallel. For each attention head, we split $X$ into non-overlapping $sh \times sw$ rectangle windows and denote the $i$-th rectangle window feature as $X_i \in \mathbb{R}^{(sh \times sw) \times C}$, where $i = 1, \ldots, \frac{H \times W}{sh \times sw}$. Then the self-attention of $X_i$ for the $m$-th head can be computed as

$$(Q_i{}^m, K_i{}^m, V_i{}^m) = (X_i W_m{}^Q, X_i W_m{}^K, X_i W_m{}^V),$$

$$Y_i^m = \text{Attention}(Q_i{}^m, K_i{}^m, V_i{}^m) = \text{SoftMax}(\frac{Q_i{}^m (K_i{}^m)^T}{\sqrt{d}} + B)V_i{}^m, \tag{1}$$

where $Y_i^m \in \mathbb{R}^{(sh \times sw) \times D}$ is the attention feature of $X_i$ in the $m$-th head, and $Q_i{}^m, K_i{}^m, V_i{}^m \in \mathbb{R}^{C \times d}$ represent the projection matrices of query, key, and value for $m$-th head. $D = d = \frac{C}{M}$ is the channel dimension in each head and $B$ is the dynamic relative position encoding [38]. Performing the attention operation on all $X_i$ ($i = 1, \ldots, \frac{H \times W}{sh \times sw}$) and reshaping and merging them in the order of division, we can get the attention feature $Y^m \in \mathbb{R}^{H \times W \times D}$ of $X$. In general, the above attention operation is the same for H-Rwin and V-Rwin, where the $sh$ and $sw$ take different values.

Assuming that the number of attention heads $M$ is even, we divide the heads equally into two parts and execute H-Rwin for the first part and V-Rwin for the second part in parallel. Finally, the outputs of two parts are concatenated along the channel dimension. The process is formulated as

$$\text{Rwin-SA}(X) = \text{Conact}(Y^1, Y^2, \ldots, Y^M)W^p, \tag{2}$$

where $Y^1, \ldots, Y^{\frac{M}{2}}$ are the output of the head using H-Rwin, and $Y^{\frac{M}{2}+1}, \ldots, Y^M$ are from the V-Rwin. The $W^p \in \mathbb{R}^{C \times C}$ represents the projection matrix for feature fusion. As shown in Fig. 3, through H-Rwin and V-Rwin, we can aggregate features cross different windows and expand the attention area without increasing computational complexity. Moreover, the rectangle window can capture different features in horizontal and vertical directions for each pixel, which is hard for square window to handle with comparable computation resource. This property is crucial for some datasets (*e.g.*, Urban100 [17]) that contain many directional and repetitive texture features.

Furthermore, as shown in Fig. 3, inspired by CSwin [12], we fix the length of one side of the rectangle to be the image resolution $H$ or $W$ (the other side length denoted as $sl$). Then the window becomes a stripe along the axis, called axial rectangle window (denoted as axial-Rwin). Since the axial window changes as the image changes, the axial-Rwin is more



Figure 3: Illustration of attention expansion, directional features aggregation, and axial-Rwin. $H$, $W$, and $sl$ mean height, width, and the other side length.

flexible than the regular rectangle window, which side length is $[sh, sw]$ or $[sw, sh]$ (denoted as regular-Rwin). The computational complexity of regular-Rwin and axial-Rwin are

$$\mathcal{O}(\text{regular-Rwin}) = HWC \times (4C + 2sh \times sw),$$
$$\mathcal{O}(\text{axial-Rwin}) = HWC \times (4C + sl \times H + sl \times W). \tag{3}$$

When $H{=}W$ (square input), the complexity of regular-Rwin is $\mathcal{O}(H^2)$ and axial-Rwin is $\mathcal{O}(H^3)$. Although axial-Rwin is more complex than regular-Rwin, it is still smaller than full-attention ($\mathcal{O}(H^4)$), and applicable for high-resolution images. And axial-Rwin has a larger attention area than that in regular-Rwin, which can capture more information, especially on the axial direction.

**Axial-Shift Operation**. To further expand the receptive field of Rwin-SA so that each pixel can aggregate more information, we propose a new shift operation called axial-shift, for Rwin-SA. As shown in Fig. 2(b), axial-shift includes two shift operations, the horizontal one is called H-Shift and the vertical one is called V-Shift, corresponding to H-Rwin and V-Rwin, respectively. The axial-shift operation moves the window partition down and left by $\frac{sh}{2}$ and $\frac{sw}{2}$ pixels, where $sh$ and $sw$ are the window height and width of H-Rwin and V-Rwin. In the implementation, we cyclically shift the feature map to down-left direction. Then we execute H-Rwin and V-Rwin on corresponding shifted feature maps and use a masking mechanism to avoid false interactions between non-adjacent pixels.

The axial-shift operation can be regarded as a more general shifted window operation in Swin Transformer [23]. When $sh$ and $sw$ in H-Rwin and V-Rwin are all the same, it degenerates into the swin shift operation. Compared with the swin shift operation, our axial-shift enables a broader range of window interaction: (1) **H(V)-Rwin with H(V)-Rwin**. Intuitively, the new window partition enables the interactive fusion of different rectangle window of the same type (horizontal or vertical one), just like the swin shift operation. (2) **H-Rwin with V-Rwin**. The Rwin-SA performs H-Rwin and V-Rwin simultaneously and fuses them through the projection in Eq. (2). For this reason, axial-shift can implicitly implement interaction between horizontal and vertical rectangle windows.

**Locality Complementary Module**. Transformer can effectively capture global information and model long-range dependencies between pixels. However, the inductive bias (translation invariance and locality) of CNN are still indispensable in image restoration tasks. It can aggregate local features and extract the basic structures of an image (*e.g.*, corners and edges). In order to complement the Transformer with the locality and achieve global and local coupling, we propose to add an independent convolution operation when calculating self-attention, called locality complementary module. As shown in Fig. 4, different from the previous practice in Transformer [42, 8], we directly operate convolution on value $V$. The process is formulated as

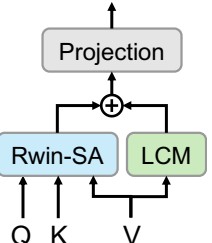

Figure 4: Illustration of locality complementary module.

$$\text{Rwin-SA}(X) = (\text{Conact}(Y^1, Y^2, \ldots, Y^M) + \text{Conv}(V))W^P, \tag{4}$$

where $Y^1, \ldots, Y^M$, and $W^P$ are the same as Eq. (2). The $V{\in}\mathbb{R}^{H \times W \times C}$ is the value projected directly from $X$ without window partition ($V_i^m$ is the partition of $V$ in $m$-th attention head defined in Eq. (1)), and $\text{Conv}(\cdot)$ represents the $3{\times}3$ depth-wise convolution.

This operation has two advantages over executing convolution sequentially or using convolution on $X$: (1) Using convolution as a parallel module allows Transformer block to adaptively choose to adopt attention or convolution operations, which is more flexible than sequential execution; (2) From Eq. (1), we can find that self-attention can be regarded as a content-dependent and dynamic weight acting on $V$. The convolution operation is equivalent to a learnable and static weight. Therefore, the convolution operation on $V$ is conducted in the same feature domain as the attention operation.

| Network | PSNR | SSIM | FLOPs |
|---|---|---|---|
| Sq. w/o shift | 32.50 | 0.9325 | 281.8G |
| Sq. w/ shift | 32.75 | 0.9347 | 281.8G |
| Re. w/o axial | 32.66 | 0.9334 | 281.8G |
| Re. w/ axial | 32.91 | 0.9360 | 281.8G |

(a) Rectangle-Window Self-Attention

| Network | PSNR | SSIM | FLOPs |
|---|---|---|---|
| C-R w/o LCM | 32.91 | 0.9360 | 281.8G |
| C-R w/ LCM | 32.98 | 0.9361 | 282.7G |
| C-A w/o LCM | 33.01 | 0.9354 | 349.7G |
| C-A w/ LCM | 33.11 | 0.9363 | 350.7G |

(b) Locality Complementary Module

| Network | PSNR | SSIM | FLOPs |
|---|---|---|---|
| C-R | 32.98 | 0.9361 | 282.7G |
| C-A-1 | 32.97 | 0.9353 | 323.5G |
| C-A-2 | 33.11 | 0.9363 | 350.7G |
| C-A-3 | 33.20 | 0.9376 | 377.9G |

(c) Window Size Impact

Table 1: Ablation study. (a) Sq.: square window ($8 \times 8$) self-attention; Re.: regular rectangle window ($4 \times 16$) self-attention; shift: swin shift operation; axial: axial-shift operation. (b) C-R: CAT-R (regular-Rwin) model, $[sw, sh]$ is $[4, 16]$ (or $16, 4$); C-A: CAT-A (axial-Rwin) model, $sl$ is $[2, 2, 2, 4, 4, 4]$; LCM: locality complementary module. (c) The $sl$ of C-A-1, C-A-2, and C-A-3 are $[2, 2, 2, 2, 2, 2]$, $[2, 2, 2, 4, 4, 4]$, and $[4, 4, 4, 4, 4, 4]$, respectively.

**Cross Aggregation Transformer Block**. CATB is composed of Rwin-SA and MLP, which has two linear projection layers with a GELU non-linearity between them. We formulate the block as

$$X' = \text{Rwin-SA}(\text{LN}(X_{in})) + X_{in},$$
$$X_{out} = \text{MLP}(\text{LN}(X')) + X', \tag{5}$$

where $X_{in}$ and $X_{out}$ are the input and output feature maps of CATB; $\text{LN}(\cdot)$ is the LayerNorm operation. And axial-shift operation is used on the interval between two consecutive cross aggregation Transformer blocks to enhance the interaction among different windows.

# 4 Experiments

## 4.1 Experimental Settings

**Data and Evaluation.** For image SR, we choose DIV2K [37] and Flickr2K [22] as the training data. And benchmark datasets for evaluation include Set5 [3], Set14 [48], B100 [27], Urban100 [17], and Manga109 [28] with three upscaling factors: $\times 2$, $\times 3$, and $\times 4$. The low-resolution images are generated by Bicubic downsampling. For JPEG compression artifact reduction, training set consists of DIV2K [37], Flickr2K [22], BSD500 [2], and WED [26]. And we have two testing datasets: Classic5 [14] and LIVE1 [35], with JPEG compression qualities of 10, 20, 30, and 40. For real image denoising, we train CAT on SIDD [1] dataset. And we have two testing datasets: SIDD validation set [1] and DND [33]. We use the metrics PSNR and SSIM [40] to evaluate the results.

**Implementation Details.** We set the residual group (RG) number as $N_1$=6 and the cross aggregation Transformer block (CATB) number as $N_2$=6 for each RG. The channel dimension, attention head number, and MLP expansion ratio for each CATB are set as 180, 6, and 4, respectively. For image SR, we build two versions of CAT, called CAT-R and CAT-A. CAT-R uses the regular rectangle window (regular-Rwin), and CAT-A uses the axial rectangle window (axial-Rwin). The regular-Rwin and the axial-Rwin are defined in Sec. 3.2. For CAT-R, we set $[sh, sw]$ as $[4, 16]$ (or $[16, 4]$) for each CATB in each RG, and for CAT-A, we set $sl$ as $[2, 2, 2, 4, 4, 4]$ for different residual groups. For JPEG compression artifact reduction, we also employ CAT-A with the same $sl$ setting as in image SR. For real image denoising, following Restormer [44], we employ a 4-level encoder-decoder. The number of Transformer blocks is $[4, 6, 6, 8]$, from level-1 to level-4. We set attention heads as $[2, 2, 4, 8]$ and MLP expansion ratio as 4. We apply axial-Rwin here and set $sl$ as 4 for all CATB.

**Training Settings.** For image SR, we train the model with batch size 32, where each input image is randomly cropped to $64 \times 64$ size, and the total training iterations are 500K. We adopt Adam optimizer [18] with $\beta_1$=0.9 and $\beta_2$=0.99 to minimize the $L_1$ loss following the work [51, 52, 21]. The initial learning rate is set as $2 \times 10^{-4}$ and reduced by half at the milestone [250K,400K,450K,475K]. For JPEG compression artifact reduction, the batch size and total iterations are 8 and 1600K, where the input image size is $128 \times 128$. Same as image SR, we adopt Adam optimizer to minimize Charbonnier Loss [4] with $\epsilon=10^{-3}$, and the learning rate is initialized to $2 \times 10^{-4}$, which is halved at [800K,1200K,1400K,1500K]. For real image denoising, we use the progressive learning proposed by Restormer [44]. The total training iterations are 300K. We adopt AdamW optimizer [25] with $\beta_1$=0.9 and $\beta_2$=0.99 to minimize the $L_1$ loss. The initial learning rate is $3 \times 10^{-4}$ and gradually reduced to $1 \times 10^{-6}$ with the cosine annealing [24]. Furthermore, we randomly utilize rotation and flipping to augment the training data for image SR, JPEG compression artifact reduction, and real image denoising. We use PyTorch [32] to implement our models with 4 Tesla V100 GPUs.[2]

---

[2] All the code and trained models are available at https://github.com/zhengchen1999/CAT.

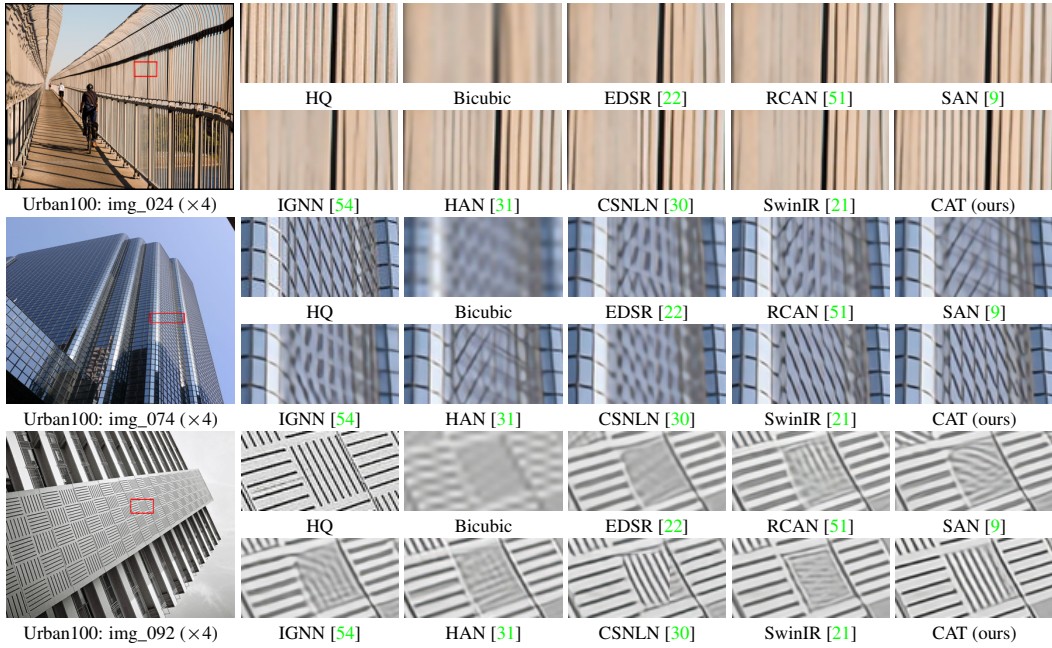

Figure 5: Visual comparison about image SR (×4) in some challenging cases.

## 4.2 Ablation Study

For the ablation study, we use the dataset DIV2K [37] and Flickr2K [22] to train CAT on the image SR (×2) task, where the input image patch size is 64×64 and the iterations are 150K. The testing dataset is Urban100 [17]. FLOPs are calculated on input image size 3×128×128.

**Rectangle-Window Self-Attention**. Table 1a demonstrates that rectangle-window self-attention is more effective than the square window attention (*e.g.*, SwinIR [21]). With no increase in computational complexity (*e.g.*, FLOPs), the rectangle window can improve the performance from 32.50 dB to 32.66 dB compared to the square one. Moreover, when we adopt the axial-shift operation in Rwin-SA, PSNR can reach 32.91 dB. And the square window attention of the swin shift operation only obtains 32.75 dB. It can be found that our Rwin-SA can aggregate more information and is more suitable for image restoration than attention mechanisms in Swin Transformer [23].

**Locality Complementary Module**. Table 1b shows the impact of locality complementary module on CAT-R (regular-Rwin) and CAT-A (axial-Rwin). With the LCM, CAT-R and CAT-A can achieve 0.07 dB and 0.10 dB gains over the cases without LCM, respectively. These results show that the inductive bias of CNN can effectively improve the performance of Transformer in image restoration. And comparing the complexity, it can be found that LCM only increases FLOPs by 0.26%~0.32%, which will not affect the model efficiency too much. Furthermore, LCM is embedded into self-attention as a new module without changing the implementation of the attention part.

**Window Size Impact**. From above analyses, we know that our Rwin-SA is more suitable for image restoration tasks than the swin attention mechanism. We further investigate the impact of different window sizes. The results are provided in Table 1c. Comparing regular-Rwin (CAT-R) and axial-Rwin (CAT-A), we can find that the performance of axial-Rwin is worse than regular-Rwin (32.97 dB vs. 32.98 dB), when the side of axial-Rwin is small ($sl$=2), despite its higher complexity (323.5G vs. 282.7G). It may be because the side of axial-Rwin ($sl$) is too narrow to capture enough information and may even introduce noise. Therefore, the result of axial-Rwin rises rapidly as $sl$ increase. These results show that both side length and window size of Rwin are crucial for performance.

## 4.3 Image Super-Resolution

We compare our two models CAT-R and CAT-A with state-of-the-art methods: EDSR [22], RCAN [51], SAN [9], IGNN [54], HAN [31], CSNLN [30], NLSA [29], IPT [5], and SwinIR [21]. Following the previous work [22, 21, 51], we use self-ensemble strategy in testing and mark the model with a symbol "+". We use CAT-A for visual comparisons, abbreviated as CAT.

| Method | Scale | Set5 | | Set14 | | B100 | | Urban100 | | Manga109 | |
|---|---|---|---|---|---|---|---|---|---|---|---|
| | | PSNR | SSIM | PSNR | SSIM | PSNR | SSIM | PSNR | SSIM | PSNR | SSIM |
| EDSR [22] | ×2 | 38.11 | 0.9602 | 33.92 | 0.9195 | 32.32 | 0.9013 | 32.93 | 0.9351 | 39.10 | 0.9773 |
| RCAN [51] | ×2 | 38.27 | 0.9614 | 34.12 | 0.9216 | 32.41 | 0.9027 | 33.34 | 0.9384 | 39.44 | 0.9786 |
| SAN [9] | ×2 | 38.31 | 0.9620 | 34.07 | 0.9213 | 32.42 | 0.9028 | 33.10 | 0.9370 | 39.32 | 0.9792 |
| IGNN [54] | ×2 | 38.24 | 0.9613 | 34.07 | 0.9217 | 32.41 | 0.9025 | 33.23 | 0.9383 | 39.35 | 0.9786 |
| HAN [31] | ×2 | 38.27 | 0.9614 | 34.16 | 0.9217 | 32.41 | 0.9027 | 33.35 | 0.9385 | 39.46 | 0.9785 |
| CSNLN [30] | ×2 | 38.28 | 0.9616 | 34.12 | 0.9223 | 32.40 | 0.9024 | 33.25 | 0.9386 | 39.37 | 0.9785 |
| NLSA [29] | ×2 | 38.34 | 0.9618 | 34.08 | 0.9231 | 32.43 | 0.9027 | 33.42 | 0.9394 | 39.59 | 0.9789 |
| IPT [5] | ×2 | 38.37 | - | 34.43 | - | 32.48 | - | 33.76 | - | - | - |
| SwinIR [21] | ×2 | 38.42 | 0.9623 | 34.46 | 0.9250 | 32.53 | 0.9041 | 33.81 | 0.9427 | 39.92 | 0.9797 |
| CAT-R (ours) | ×2 | 38.48 | 0.9625 | 34.53 | 0.9251 | 32.56 | 0.9045 | 34.08 | 0.9443 | 40.09 | 0.9804 |
| CAT-A (ours) | ×2 | 38.51 | 0.9626 | 34.78 | 0.9265 | 32.59 | 0.9047 | 34.26 | 0.9440 | 40.10 | 0.9805 |
| CAT-R+ (ours) | ×2 | 38.52 | 0.9627 | 34.59 | 0.9257 | 32.58 | 0.9047 | 34.19 | 0.9450 | 40.18 | 0.9805 |
| CAT-A+ (ours) | ×2 | 38.55 | 0.9628 | 34.81 | 0.9267 | 32.60 | 0.9048 | 34.34 | 0.9445 | 40.18 | 0.9806 |
| EDSR [22] | ×3 | 34.65 | 0.9280 | 30.52 | 0.8462 | 29.25 | 0.8093 | 28.80 | 0.8653 | 34.17 | 0.9476 |
| RCAN [51] | ×3 | 34.74 | 0.9299 | 30.65 | 0.8482 | 29.32 | 0.8111 | 29.09 | 0.8702 | 34.44 | 0.9499 |
| SAN [9] | ×3 | 34.75 | 0.9300 | 30.59 | 0.8476 | 29.33 | 0.8112 | 28.93 | 0.8671 | 34.30 | 0.9494 |
| IGNN [54] | ×3 | 34.72 | 0.9298 | 30.66 | 0.8484 | 29.31 | 0.8105 | 29.03 | 0.8696 | 34.39 | 0.9496 |
| HAN [31] | ×3 | 34.75 | 0.9299 | 30.67 | 0.8483 | 29.32 | 0.8110 | 29.10 | 0.8705 | 34.48 | 0.9500 |
| CSNLN [30] | ×3 | 34.74 | 0.9300 | 30.66 | 0.8482 | 29.33 | 0.8105 | 29.13 | 0.8712 | 34.45 | 0.9502 |
| NLSA [29] | ×3 | 34.85 | 0.9306 | 30.70 | 0.8485 | 29.34 | 0.8117 | 29.25 | 0.8726 | 34.57 | 0.9508 |
| IPT [5] | ×3 | 34.81 | - | 30.85 | - | 29.38 | - | 29.49 | - | - | - |
| SwinIR [21] | ×3 | 34.97 | 0.9318 | 30.93 | 0.8534 | 29.46 | 0.8145 | 29.75 | 0.8826 | 35.12 | 0.9537 |
| CAT-R (ours) | ×3 | 34.99 | 0.9320 | 31.00 | 0.8539 | 29.49 | 0.8154 | 29.91 | 0.8848 | 35.29 | 0.9542 |
| CAT-A (ours) | ×3 | 35.06 | 0.9326 | 31.04 | 0.8538 | 29.52 | 0.8160 | 30.12 | 0.8862 | 35.38 | 0.9546 |
| CAT-R+ (ours) | ×3 | 35.07 | 0.9324 | 31.06 | 0.8544 | 29.52 | 0.8159 | 30.05 | 0.8864 | 35.44 | 0.9548 |
| CAT-A+ (ours) | ×3 | 35.10 | 0.9327 | 31.09 | 0.8545 | 29.55 | 0.8164 | 30.21 | 0.8872 | 35.48 | 0.9550 |
| EDSR [22] | ×4 | 32.46 | 0.8968 | 28.80 | 0.7876 | 27.71 | 0.7420 | 26.64 | 0.8033 | 31.02 | 0.9148 |
| RCAN [51] | ×4 | 32.63 | 0.9002 | 28.87 | 0.7889 | 27.77 | 0.7436 | 26.82 | 0.8087 | 31.22 | 0.9173 |
| SAN [9] | ×4 | 32.64 | 0.9003 | 28.92 | 0.7888 | 27.78 | 0.7436 | 26.79 | 0.8068 | 31.18 | 0.9169 |
| IGNN [54] | ×4 | 32.57 | 0.8998 | 28.85 | 0.7891 | 27.77 | 0.7434 | 26.84 | 0.8090 | 31.28 | 0.9182 |
| HAN [31] | ×4 | 32.64 | 0.9002 | 28.90 | 0.7890 | 27.80 | 0.7442 | 26.85 | 0.8094 | 31.42 | 0.9177 |
| CSNLN [30] | ×4 | 32.68 | 0.9004 | 28.95 | 0.7888 | 27.80 | 0.7439 | 27.22 | 0.8168 | 31.43 | 0.9201 |
| NLSA [29] | ×4 | 32.59 | 0.9000 | 28.87 | 0.7891 | 27.78 | 0.7444 | 26.96 | 0.8109 | 31.27 | 0.9184 |
| IPT [5] | ×4 | 32.64 | - | 29.01 | - | 27.82 | - | 27.26 | - | - | - |
| SwinIR [21] | ×4 | 32.92 | 0.9044 | 29.09 | 0.7950 | 27.92 | 0.7489 | 27.45 | 0.8254 | 32.03 | 0.9260 |
| CAT-R (ours) | ×4 | 32.89 | 0.9044 | 29.13 | 0.7955 | 27.95 | 0.7500 | 27.62 | 0.8292 | 32.16 | 0.9269 |
| CAT-A (ours) | ×4 | 33.08 | 0.9052 | 29.18 | 0.7960 | 27.99 | 0.7510 | 27.89 | 0.8339 | 32.39 | 0.9285 |
| CAT-R+ (ours) | ×4 | 32.98 | 0.9049 | 29.18 | 0.7963 | 27.98 | 0.7506 | 27.73 | 0.8310 | 32.35 | 0.9280 |
| CAT-A+ (ours) | ×4 | 33.14 | 0.9059 | 29.23 | 0.7968 | 28.01 | 0.7516 | 27.99 | 0.8356 | 32.52 | 0.9293 |

Table 2: Quantitative comparison (PSNR/SSIM) with state-of-the-art methods for image SR. Best and second best results are colored with red and blue.

| Dataset | q | RNAN [52] | | RDN [53] | | DRUNet [49] | | SwinIR [21] | | CAT (ours) | | CAT+ (ours) | |
|---|---|---|---|---|---|---|---|---|---|---|---|---|---|
| | | PSNR | SSIM | PSNR | SSIM | PSNR | SSIM | PSNR | SSIM | PSNR | SSIM | PSNR | SSIM |
| LIVE1 | 10 | 29.63 | 0.8239 | 29.67 | 0.8247 | 29.79 | 0.8278 | 29.86 | 0.8287 | 29.89 | 0.8295 | 29.92 | 0.8299 |
| | 20 | 32.03 | 0.8877 | 32.07 | 0.8882 | 32.17 | 0.8899 | 32.25 | 0.8909 | 32.30 | 0.8913 | 32.32 | 0.8915 |
| | 30 | 33.45 | 0.9149 | 33.51 | 0.9153 | 33.59 | 0.9166 | 33.69 | 0.9174 | 33.73 | 0.9177 | 33.75 | 0.9179 |
| | 40 | 34.47 | 0.9299 | 34.51 | 0.9302 | 34.58 | 0.9312 | 34.67 | 0.9317 | 34.72 | 0.9320 | 34.74 | 0.9322 |
| Classic5 | 10 | 29.96 | 0.8178 | 30.00 | 0.8188 | 30.16 | 0.8234 | 30.27 | 0.8249 | 30.26 | 0.8250 | 30.30 | 0.8257 |
| | 20 | 32.11 | 0.8693 | 32.15 | 0.8699 | 32.39 | 0.8734 | 32.52 | 0.8748 | 32.57 | 0.8754 | 32.60 | 0.8756 |
| | 30 | 33.38 | 0.8924 | 33.43 | 0.8930 | 33.59 | 0.8949 | 33.73 | 0.8961 | 33.77 | 0.8964 | 33.80 | 0.8966 |
| | 40 | 34.27 | 0.9061 | 34.27 | 0.9061 | 34.41 | 0.9075 | 34.52 | 0.9082 | 34.58 | 0.9087 | 34.60 | 0.9088 |

Table 3: Quantitative comparison (PSNR/SSIM) with state-of-the-art methods for JPEG compression artifact reduction. Best and second best results are colored with red and blue.

**Quantitative Results.** Table 2 shows PSNR/SSIM comparisons for ×2, ×3, and ×4 image SR. As we can see, our CAT-R (regular-Rwin) and CAT-A (axial-Rwin) significantly outperform other methods on all datasets with all scale factors. Among these five datasets, CAT-R and CAT-A have the most noticeable improvement in Urban100. Compared with the recent Transformer model (*e.g.*, SwinIR), for CAT-R, the PSNR gain reaches 0.27 dB for scale factor 2, and for CAT-A, the PSNR increases by 0.45 dB. It shows that our CAT can capture more global information than previous CNN-based and Transformer-based models, which is very effective for images in Urban100 with a large number of repetitive texture structures. Moreover, compared with CAT-R, CAT-A yields 0.18~0.27 dB improvements on Urban100. All these results indicate the effectiveness of our method.

**Visual Results.** We show visual comparisons (×4) in Fig. 5. We can observe that most compared methods can hardly recover accurate textures and suffer from blurring artifacts in some challenging cases. In contrast, our CAT can reconstruct more high-frequency structural details and alleviate the blurring artifacts. For example, in img_092, our CAT can recover correct and sharp textures, while

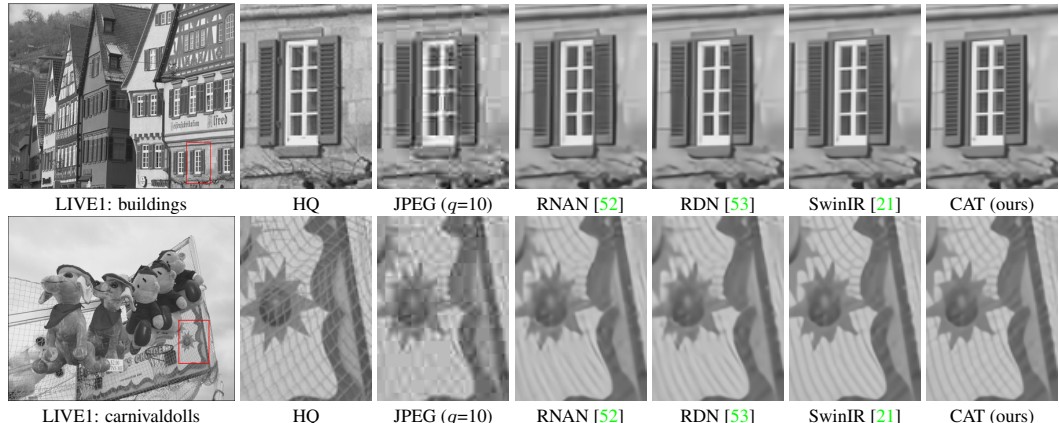

Figure 6: Visual comparison about JPEG compression artifacts reduction ($q$=10).

| Dataset | | DANet+ [43] | CycleISP [45] | MIRNet [46] | MPRNet [47] | Uformer [39] | Restormer [44] | CAT (ours) | CAT+ (ours) |
|---|---|---|---|---|---|---|---|---|---|
| Parameters (M) | | 9.15 | 2.83 | 31.79 | 15.74 | 50.88 | 26.11 | 25.77 | 25.77 |
| SIDD* | PSNR | 39.47 | 39.52 | 39.72 | 39.71 | 39.89 | 40.02 | 40.01 | 40.05 |
| | SSIM | 0.9570 | 0.9571 | 0.9586 | 0.9586 | 0.9594 | 0.9603 | 0.9600 | 0.9602 |
| DND | PSNR | 39.58 | 39.56 | 39.88 | 39.82 | 39.98 | 40.03 | 40.05 | 40.08 |
| | SSIM | 0.9545 | 0.9564 | 0.9563 | 0.9540 | 0.9554 | 0.9564 | 0.9561 | 0.9563 |

Table 4: Quantitative comparison (PSNR/SSIM) for real image denoising. Best and second best results are colored with red and blue. * We re-test the SIDD with all official pre-trained models.

most compared methods fail to recover image details. These results demonstrate that our CAT has more robust representational ability to recover structural contents and textural details. It also shows that, compared with other models, our CAT is more effective in modeling long-range dependencies, which are crucial for reconstructing high-quality images.

## 4.4 JPEG Compression Artifact Reduction

We compare our CAT with state-of-the-art image compression artifact removal methods: RNAN [52], RDN [53], DRUNet [49], and SwinIR [21]. Here, we only focus on the restoration of Y channel (in YCbCr space). We also use self-ensemble strategy and mark the model with a symbol "+". Quantitative and visual comparisons are shown in Table 3 and Fig. 6, respectively.

**Quantitative Results.** Table 3 shows quantitative comparisons with four JPEG quality settings: 10, 20, 30, and 40. Our CAT+ performs better than other compared methods on LIVE1 and Classic5 with all JPEG qualities. Even without self-ensemble, our RCAN also outperforms other compared methods, except for PSNR on Classic5 ($q$=10). Especially when $q$=40, CAT achieves 0.05 and 0.06 dB improvements over the Transformer-based model SwinIR. Even in a very low compression quality (*e.g.*, 10), CAT achieves 29.89 dB in LIVE1, higher than previous methods.

**Visual Results.** We show visual comparisons at very low image quality ($q$=10) in Fig. 6. For previous methods, RNAN [52], RDN [53], and SwinIR [21], blocking artifacts are hardly to be totally removed and some structures are over-smoothed in some challenging cases. In contrast, our CAT can recover more details and remove blocking artifacts, resulting in sharper edges and more explicit textures. These visual comparisons further demonstrate the effectiveness of our proposed CAT.

## 4.5 Real Image Denoising

We further conduct real image denoising experiments. We compare our CAT with state-of-the-art methods: DANet+ [43], CycleISP [45], MIRNet [46], MPRNet [47], Uformer [39], and Restormer [44]. We use self-ensemble [22] strategy and mark the model with a symbol "+".

**Quantitative Results.** Table 4 shows quantitative comparisons for real image denoising. We re-test the SIDD results with all official pre-trained models on previous state-of-the-art methods. Our CAT performs better than other compared methods on SIDD and DND, except Restormer [44]. Meanwhile, our method has comparable performance to Restormer with fewer parameters. Our CAT achieves 0.02 dB improvements over Restormer on DND. All these show the superiority of our method.

| Method | EDSR [22] | RCAN [51] | HAN [31] | CSNLN [30] | SwinIR [21] | CAT-R (ours) | CAT-A (ours) | CAT-R-2 (ours) |
|---|---|---|---|---|---|---|---|---|
| PSNR (dB) | 26.64 | 26.82 | 26.85 | 27.22 | 27.45 | 27.62 | 27.89 | 27.59 |
| FLOPs (G) | 823.3 | 261.0 | 269.1 | 84,155.2 | 215.3 | 292.7 | 360.7 | 216.3 |
| Parameters (M) | 43.09 | 15.59 | 16.07 | 6.57 | 11.90 | 16.60 | 16.60 | 11.93 |

Table 5: Model complexity comparisons ($\times 4$). Output size is $3 \times 512 \times 512$ to calculate FLOPs.

### 4.6 Model Size Analyses

Table 5 shows the comparison of performance, computational complexity (*e.g.*, FLOPs), and parameter numbers on image SR. FLOPs are measured when output size is set to $3 \times 512 \times 512$ and PSNR values are tested on Urban100 ($\times 4$). Although our CAT achieves excellent performance, it has less computational complexity and parameters than EDSR [22]. And the computational complexity of our CAT is much lower than CSNLN [30]. Compared to other CNN-based and Transformer-based methods, our CAT still has a comparable complexity and model size. To demonstrate the effectiveness of our method, we provide another variant of CAT (CAT-R-2) with similar computational complexity and parameters to SwinIR. Our CAT-R-2 still outperforms SwinIR and other state-of-the-art methods. More details about CAT-R-2 are provided in the supplementary material.

## 5 Conclusion

In this paper, we propose a new Transformer model named cross aggregation Transformer (CAT) for image restoration with the rectangle-window self-attention (Rwin-SA) mechanism as the key part. Specifically, Rwin-SA performs H-Rwin and V-Rwin attention parallelly in different heads. This mechanism can aggregate the features cross different windows and increase the receptive field without additional computational cost. Meanwhile, we propose an axial-shift operation according to the property of rectangle-window to further fuse different windows, which complements Rwin-SA. Furthermore, the locality complementary module (LCM) introduces locality into self-attention, adding the inductive bias of CNN to Transformer. Through the usage of LCM, Rwin-SA can realize the coupling of global and local information, which is more conducive to image restoration. Extensive experiments on image SR, JPEG compression artifact reduction, and real image denoising demonstrate that our proposed CAT outperforms current state-of-the-art methods.

**Acknowledgments**. This work is partly supported by NSFC (62141220, 61972253, U1908212, 72061127001), the Program for Professor of Special Appointment (Eastern Scholar) at Shanghai Institutions of Higher Learning. Xin Yuan acknowledges the support of NSFC (62271414), Westlake Foundation (2021B1501-2) and the Research Center for Industries of the Future (RCIF) at Westlake University.

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
