# Supplementary Material: Cross Aggregation Transformer for Image Restoration

**Zheng Chen[1], Yulun Zhang[2], Jinjin Gu[3,4], Yongbing Zhang[5], Linghe Kong[1]\*, Xin Yuan[6]**
[1]Shanghai Jiao Tong University, [2]ETH Zürich, [3]Shanghai AI Laboratory,
[4]The University of Sydney, [5]Harbin Institute of Technology (Shenzhen), [6]Westlake University

## 1 Method

### 1.1 Source Code

We provide the source code and pretrained models at `https://github.com/zhengchen1999/CAT`.

### 1.2 Variant Models

We provide two variant models for image SR, called CAT-R-2 and CAT-A-2. For two models, we set residual group (RG) numbers $N_1$, cross aggregation Transformer block (CATB) number $N_2$, channel dimension, and attention head number as 6, 6, 180, and 6, respectively. These settings are consistent with CAT-R and CAT-A. For CAT-R-2, we apply regular-Rwin, and set $[sw, sh]$ as $[4, 16]$ (same as CAT-R). We set the MLP expansion ratio as 2, consistent with SwinIR [13]. For CAT-A-2, we apply axial-Rwin, and set $sl$ as 4 for all CATB in each RG. The MLP expansion ratio is set as 4.

| Method | Scale | Set5 | | Set14 | | B100 | | Urban100 | | Manga109 | |
|---|---|---|---|---|---|---|---|---|---|---|---|
| | | PSNR | SSIM | PSNR | SSIM | PSNR | SSIM | PSNR | SSIM | PSNR | SSIM |
| SwinIR [13] | ×2 | 38.42 | 0.9623 | 34.46 | 0.9250 | 32.53 | 0.9041 | 33.81 | 0.9427 | 39.92 | 0.9797 |
| CAT-R-2 | ×2 | 38.47 | 0.9624 | 34.51 | 0.9251 | 32.55 | 0.9043 | 34.03 | 0.9439 | 40.05 | 0.9802 |
| CAT-A-2 | ×2 | 38.56 | 0.9628 | 34.78 | 0.9267 | 32.59 | 0.9047 | 34.38 | 0.9452 | 40.14 | 0.9805 |
| CAT-R-2+ | ×2 | 38.52 | 0.9626 | 34.56 | 0.9257 | 32.58 | 0.9046 | 34.15 | 0.9446 | 40.15 | 0.9804 |
| CAT-A-2+ | ×2 | 38.58 | 0.9629 | 34.83 | 0.9269 | 32.61 | 0.9049 | 34.46 | 0.9457 | 40.21 | 0.9807 |
| SwinIR [13] | ×3 | 34.97 | 0.9318 | 30.93 | 0.8534 | 29.46 | 0.8145 | 29.75 | 0.8826 | 35.12 | 0.9537 |
| CAT-R-2 | ×3 | 35.03 | 0.9321 | 30.97 | 0.8534 | 29.47 | 0.8150 | 29.85 | 0.8838 | 35.25 | 0.9540 |
| CAT-A-2 | ×3 | 35.09 | 0.9327 | 31.09 | 0.8545 | 29.53 | 0.8162 | 30.22 | 0.8882 | 35.44 | 0.9549 |
| CAT-R-2+ | ×3 | 35.08 | 0.9323 | 31.03 | 0.8542 | 29.50 | 0.8155 | 30.01 | 0.8855 | 35.41 | 0.9546 |
| CAT-A-2+ | ×3 | 35.14 | 0.9329 | 31.13 | 0.8549 | 29.55 | 0.8165 | 30.32 | 0.8892 | 35.55 | 0.9552 |
| SwinIR [13] | ×4 | 32.92 | 0.9044 | 29.09 | 0.7950 | 27.92 | 0.7489 | 27.45 | 0.8254 | 32.03 | 0.9260 |
| CAT-R-2 | ×4 | 32.91 | 0.9040 | 29.13 | 0.7953 | 27.93 | 0.7493 | 27.59 | 0.8285 | 32.16 | 0.9263 |
| CAT-A-2 | ×4 | 33.09 | 0.9054 | 29.21 | 0.7964 | 27.99 | 0.7513 | 27.99 | 0.8357 | 32.47 | 0.9290 |
| CAT-R-2+ | ×4 | 32.97 | 0.9048 | 29.20 | 0.7962 | 27.97 | 0.7499 | 27.71 | 0.8306 | 32.34 | 0.9276 |
| CAT-A-2+ | ×4 | 33.12 | 0.9057 | 29.26 | 0.7972 | 28.02 | 0.7518 | 28.08 | 0.8371 | 32.61 | 0.9298 |

Table 1: Quantitative comparison (PSNR/SSIM) with SwinIR [13] for image SR. Best and second best results are colored with red and blue.

### 1.3 Quantitative Results

We train CAT-R-2 and CAT-A-2 on DIV2K [26] and Flickr2K [14] in the same way (training settings) we train CAT-R and CAT-A. We test two models on Set5 [2], Set14 [27], B100 [20], Urban100 [11], and Manga109 [21] with three upscaling factors: ×2, ×3, and ×4. We compare two variants with SwinIR. We use self-ensemble strategy and mark models with "+". The results are shown in Table 1.

As we can see, our CAT-A-2 significantly outperforms SwinIR [13] on all datasets with all scale factors. And CAT-A-2 still performs better than SwinIR, except for Se5 (×4). CAT-A-2 achieves 0.57 dB gain over SwinIR on Urban100 (×2), and 0.44 dB gain on Manga109 (×2). Moreover, our

---

\*Corresponding author: Linghe Kong, linghe.kong@sjtu.edu.cn

| Method | Params (M) | FLOPs (G) | Set5 | Set14 | B100 | Urban100 | Manga109 |
|---|---|---|---|---|---|---|---|
| SwinIR [13] | 11.90 | 215.3 | 32.92 | 29.09 | 27.92 | 27.45 | 32.03 |
| CAT-R-2 | 11.93 | 216.3 | 32.91 | 29.13 | 27.93 | 27.59 | 32.16 |
| CAT-A-2 | 16.60 | 387.9 | 33.09 | 29.21 | 27.99 | 27.99 | 32.47 |

Table 2: Model complexity comparisons (×4). Output size is 3×512×512 to calculate FLOPs.

CAT-R-2 achieves 0.22 dB on Urban100 (×2), at similar computational complexity to SwinIR. We will discuss this in detail in Sec. 1.4. In addition, compared with CAT-A, the variant model CAT-A-2 yields 0.1~0.12 dB gains. All these results further indicate the effectiveness of our method.

## 1.4 Model Size Analyses

Table 2 shows the comparison of performance, computational complexity (*e.g.*, FLOPs), and parameter numbers on image SR. FLOPs are measured when the output size is set to 3×512×512. Our CAT-R-2 has similar parameters and complexity of SwinIR [13]. The parameter numbers and computational complexity only increase by 0.25% and 0.46%, respectively. From the main paper, we can know that the extra parameters and complexity come from locality complementary module (LCM), which is crucial to performance. With a slight increase in complexity, our CAT-R-2 achieves 0.14 dB and 0.13 dB on Urban100 and Manga109, respectively. And CAT-R-2 outperforms SwinIR on other benchmark datasets, except for Set5. For CAT-A-2, it has the same number of parameters as CAT-R and CAT-A. With a slight increase in complexity, our CAT-A-2 can significantly improve performance. CAT-A-2 obtains 0.1 dB boost over CAT-A.

## 2 Experimental Results

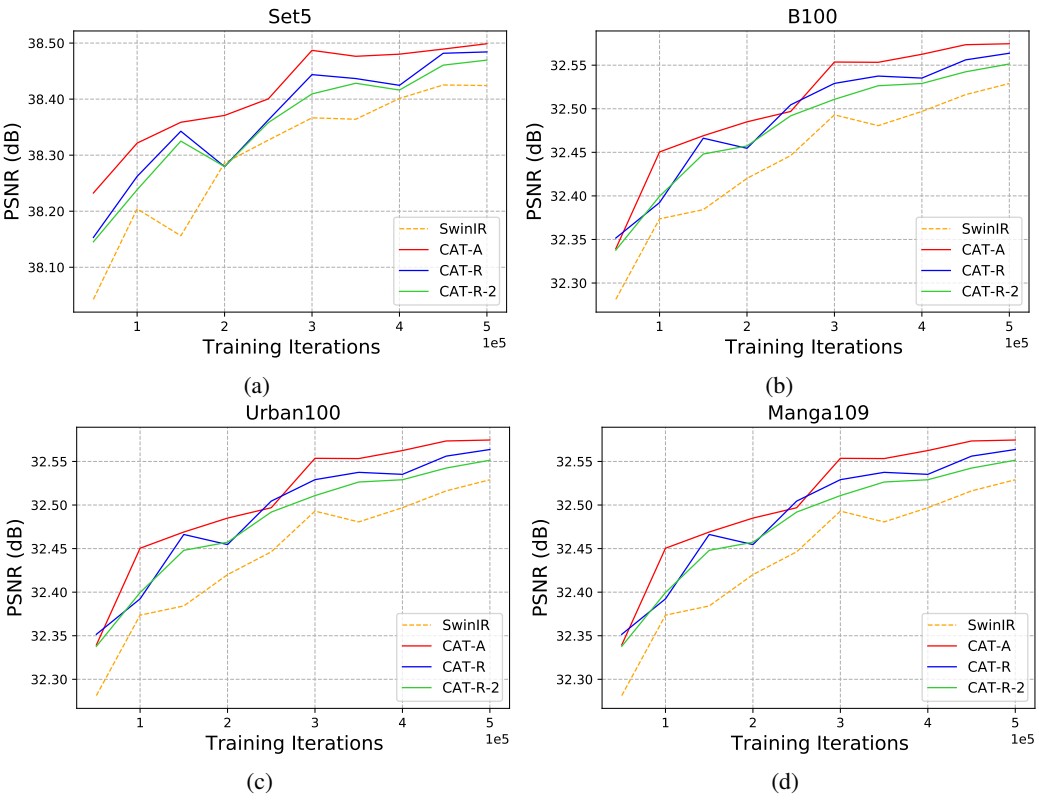

Figure 1: Convergence analyses on CAT-A, CAT-R, CAT-R-2, and SwinIR [13].

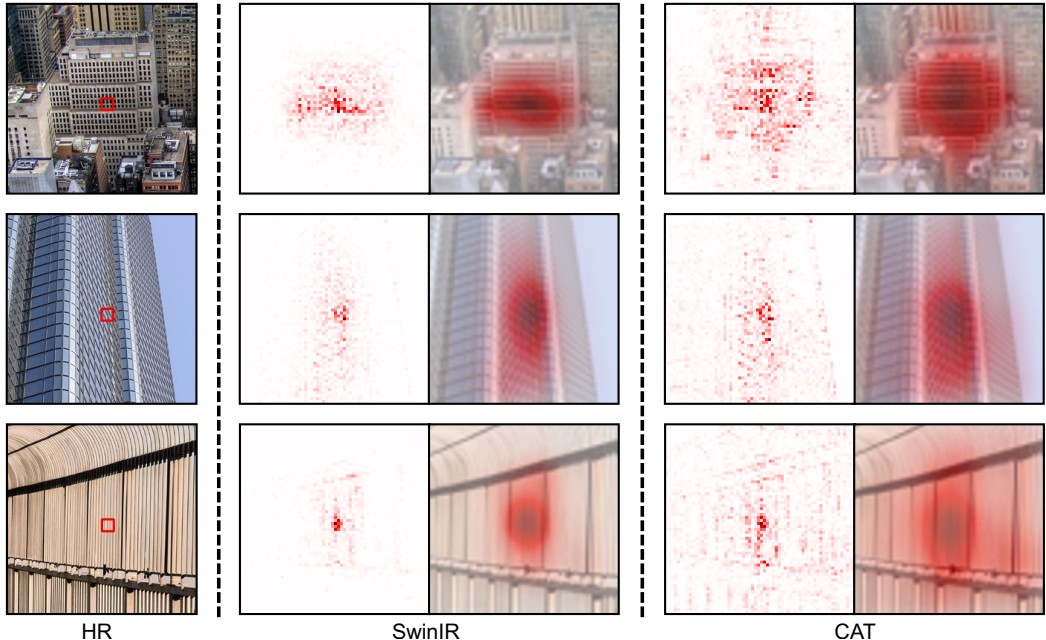

HR              SwinIR             CAT

Figure 2: LAM [8] comparison between SwinIR [13] and CAT.

## 2.1 Convergence Analyses

We plot the PSNR during training for SwinIR, CAT-A, CAT-R, and CAT-R-2 in Fig. 1. PSNR values are tested on Set5 [2], Set14 [27], B100 [20], Urban100 [11], and Manga109 [21] for image SR (×2). We can observe that our CAT-A, CAT-R and CAT-R-2 convergence is faster than SwinIR on all datasets. For CAT-R-2, it has a similar convergence speed as CAT-R, albeit with less computational complexity and parameters. Moreover, CAT-A converges much faster and better than other models. It indicates the effectiveness of our axial-Rwin self-attention mechanism. All these results demonstrate the superior performance of our proposed cross aggregation Transformer (CAT).

## 2.2 LAM Analyses

We use LAM [8] to visualize the receptive fields of CAT and SwinIR [13]. LAM is an attribution method designed for SR, which can show pixels that contribute most to the SR result. In other words, the more pixels that can be utilized, the larger the actual receptive field of the model. We display three sets of comparison plots in Fig. 2. We can observe that SwinIR can only utilize a limited range of pixels. In contrast, our CAT has a global receptive field, in which available pixels are extended to almost complete images. All these results show that our CAT can capture global information and have long-range modeling ability. Furthermore, these visualization results are consistent with the quantitative and visual comparison in Table 7, Figs. 4, 5, and 6.

## 2.3 Image Super-Resolution

We compare our method with 20 state-of-the-art methods: EDSR [14], D-DBPN [9], SRMDNF [28], RDN [31], OISR [10], RCAN [29], NLRN [15], RNAN [30], SRFBN [12], SAN [4], RFANet [16], NSR [6], IGNN [33], HAN [24], CSNLN [23], NLSA [22], CRAN [32], DFSA+ [19], IPT [3], and SwinIR [13]. We use self-ensemble strategy in testing and mark the model with a symbol "+". We use CAT-A for visual comparisons, abbreviated as CAT. Quantitative comparisons are shown in Table 7. Visual comparisons are shown in Figs. 4, 5, and 6.

**Quantitative Comparisons**. Table 7 shows more PSNR/SSIM comparisons for ×2, ×3, and ×4 image SR. Our AT-R (regular-Rwin) and CAT-A (axial-Rwin) significantly outperform other methods on all datasets with all scale factors. All these results indicate the effectiveness of our method.

**Visual Comparisons**. We provide more visual comparisons in Figs. 4, 5, and 6. For example, in img_011, our CAT can recover the lines completely, while most compared methods fail to recover

| Method | q=10 | | q=20 | | q=30 | | q=40 | |
|---|---|---|---|---|---|---|---|---|
| | PSNR | SSIM | PSNR | SSIM | PSNR | SSIM | PSNR | SSIM |
| SwinIR [13] | 30.55 | 0.8841 | 33.12 | 0.9252 | 34.58 | 0.9418 | 35.50 | 0.9508 |
| CAT | 30.80 | 0.8875 | 33.38 | 0.9274 | 34.81 | 0.9432 | 35.73 | 0.9520 |
| CAT+ | 30.89 | 0.8885 | 33.46 | 0.9280 | 34.88 | 0.9436 | 35.81 | 0.9523 |

Table 3: Quantitative comparison (PSNR/SSIM) on Urban100 with SwinIR [13] for JPEG compression artifact reduction. Best and second best results are colored with red and blue.

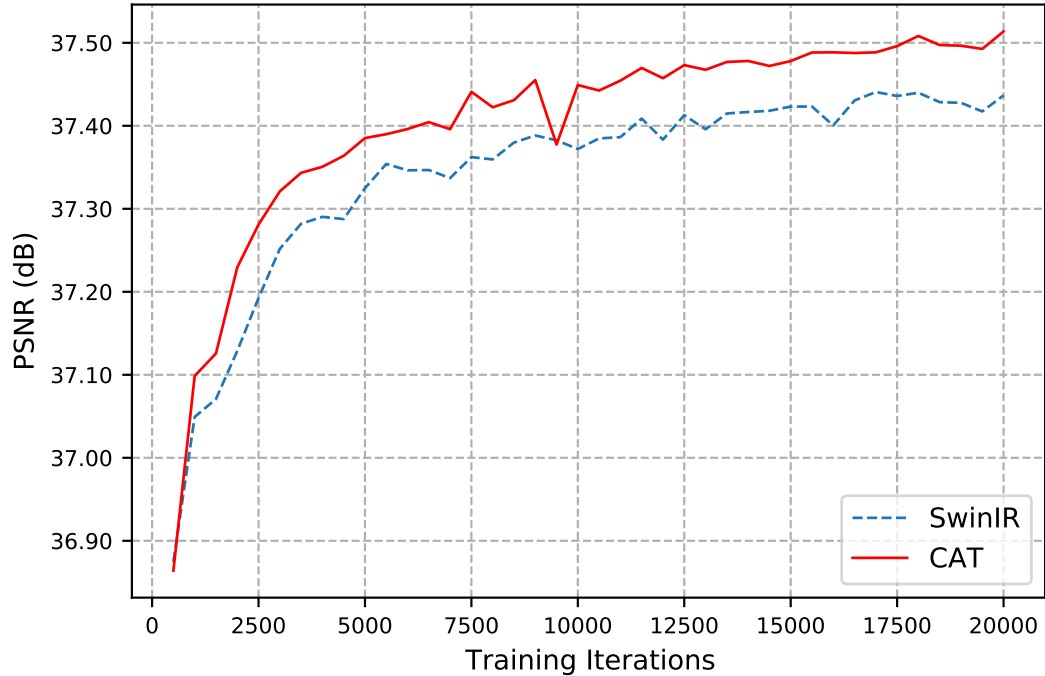

Figure 3: Convergence analyses on CAT and SwinIR on color JPEG compression artifact reduction.

lines near the bottom. In img_048, compared methods cannot recover textures at the top of the pyramid architecture. In contrast, our CAT can recover right and sharp lattices. In LoveHina_vol01, our CAT can alleviate the blurring artifacts better and recover the girl's hair, while other methods suffer from blurring artifacts. These visual comparisons are consistent with the quantitative results and demonstrate the effectiveness of our method with the usage of rectangle-window attention .

### 2.4 Grayscale JPEG Compression Artifact Reduction

Our CAT has a more robust representational ability to recover structural contents and texture details due to rectangle window self-attention. However, for the JPEG artifact reduction testing datasets: Classic5 [7] and LIVE1 [25], the number of images they contain is small (5 and 29), and the texture features are not rich. So the overall improvement effect is not obvious.

To demonstrate the effectiveness of our method, we further compare our CAT with SwinIR [13] on Urban100 [11] with JPEG compression qualities of 10, 20, 30, and 40. Here, we focus on the restoration of Y channel (in YCbCr space). We still use self-ensemble strategy and mark the model with a symbol "+". Quantitative and visual comparisons are shown in Table 3, Figs. 7 and 8.

**Quantitative Comparisons**. Table 3 shows quantitative comparisons with SwinIR [13] on Urban100. Our CAT significantly outperforms SwinIR. Unlike the slight increase on Classic5 [7] and LIVE1 [25] (0.06 dB), our CAT+ yields 0.30~0.34 dB gains on Urban100. Even without self-ensemble, our CAT also achieves 0.25~0.26 dB gains. These results show that our CAT can capture more global information than SwinIR, which is crucial to images with directional and repetitive texture features.

**Visual Comparisons**. We provide more visual comparisons in Figs. 7 and 8. We only compare our CAT with SwinIR on Urban100. For example, in img_019, we can observe that our CAT can recover more details and remove blocking artifacts, while SwinIR restores some wrong textures. In

| Method | Iteration | 25K | 50K | 75K | 100K | 125K | 150K | 175K | 200K |
|--------|-----------|-----|-----|-----|------|------|------|------|------|
| SwinIR | PSNR | 34.73 | 34.86 | 34.92 | 34.95 | 34.97 | 34.99 | 35.00 | 35.01 |
|        | SSIM | 0.9347 | 0.9359 | 0.9364 | 0.9368 | 0.9370 | 0.9372 | 0.9372 | 0.9370 |
| CAT (ours) | PSNR | **34.82** | **34.95** | **35.02** | **35.05** | **35.07** | **35.08** | **35.10** | **35.11** |
|            | SSIM | **0.9354** | **0.9367** | **0.9374** | **0.9378** | **0.9379** | **0.9379** | **0.9381** | **0.9380** |

Table 4: Quantitative comparison (PSNR/SSIM) on LIVE1 with SwinIR [13] for color JPEG compression artifact reduction ($q$=40) on checkpoints, from 0 to 200K (iterations).

| Method | Set5 | | Set14 | | LIVE1 | | Urban100 | |
|--------|------|------|-------|------|-------|------|----------|------|
|        | PSNR | SSIM | PSNR | SSIM | PSNR | SSIM | PSNR | SSIM |
| SwinIR [13] | 37.44 | 0.9487 | 35.74 | 0.9319 | 35.01 | 0.9370 | 35.42 | 0.9520 |
| CAT (ours) | **37.51** | **0.9491** | **35.87** | **0.9326** | **35.11** | **0.9380** | **35.76** | **0.9539** |

Table 5: Quantitative comparison (PSNR/SSIM) with SwinIR [13] for color JPEG compression artifact reduction ($q$=40). Training iterations are 200K for both CAT and SwinIR.

img_060, SwinIR cannot recover correct letters and over-smooth some textures. In contrast, our CAT can restore explicit letters and textures. In img_074, our CAT can recover the lattices in high places, while SwinIR suffers from blurring artifacts. These results demonstrate that our CAT has the more powerful long-range dependencies modeling ability and can capture more global information.

## 2.5 Color JPEG Compression Artifact Reduction

We further compare our CAT with SwinIR [13] on color JPEG compression artifact reduction. The convergence analyses are in Fig. 3, and quantitative comparisons are in Tables 4 and 5.

**Experimental Settings**. We still use the CAT for (grayscale) JPEG compression artifact reduction we proposed in the main paper. We change the input and output channels from 1 to 3. SwinIR has the same modification. The training setting is still the same as (grayscale) JPEG compression artifact reduction task. More details are shown in the main paper.

We train CAT and SwinIR on DIV2K [26], Flickr2K [14], BSD500 [1], and WED [18]. And we have four testing datasets: Set5 [2], Set14 [27], LIVE1 [25], and Urban100 [11], with JPEG compression qualities of 40. We calculate PSNR and SSIM [18] on the Y channel of the YCbCr space.

**Convergence Analyses**. Due to time issues, we only completed part of the training (fininshed iterations = 200K, target total iterations = 1600K.). In Fig. 3, we show the validation curves of our CAT and SwinIR during training, from 0 to 200K (iterations). PSNR values are tested on Set5 [2]. We can observe that our CAT convergence is faster than SwinIR.

**Quantitative Comparisons**. Table 4 shows the comparisons of the performance on LIVE1 of CAT and SwinIR during training, from 0 to 200K (iterations). Our CAT outperforms SwinIR on all checkpoints. And Table 5 shows quantitative comparisons with SwinIR when the iterations are 200K. Our CAT outperforms SwinIR on all datasets. Our CAT yields 0.1 dB gains on Urban100 and 0.34 dB gains on LIVE1. These results demonstrate the effectiveness of our CAT.

## 2.6 Other Numerical Results

| Method | Params (M) | FLOPs (G) | Set5 | Set14 | B100 | Urban100 | Manga109 |
|--------|-----------|-----------|------|-------|------|----------|----------|
| CSwin [5] | 16.45 | 350.7 | 38.40 | 34.42 | 32.46 | 33.73 | 39.83 |
| CAT-A | 16.46 | 350.7 | **38.51** | **34.78** | **32.59** | **34.26** | **40.10** |

Table 6: Model comparisons (×2). Output size is 3×256×256 to calculate FLOPs.

**CAT vs. CSwin**. To demonstrate the superiority of our CAT, we compare the performance of CAT-A and CSwin [5] on image SR (×2). The CSwin model uses our CAT architecture and replaces our Cross Aggregation Transformer Block (CATB) with the CSWin Transformer Block. The implementation details and training settings are the same for CAT and CSwin. Table 6 shows the comparison of performance, computational complexity (*e.g.*, FLOPs), and parameter numbers on image SR (×2). FLOPs are measured when the output size is set to 3×256×256. Our CAT-A significantly outperforms CSwin on all datasets with similar model sizes and computational complexity.

# 3 Additional Analyses

**Difference between axial-shift and shift operation in Swin Transformer**. Our axial-shift reference the design of the shift operation in Swin Transformer. However, the axial-shift we proposed is different from the shifted window operation in Swin Transformer [17].

The most significant difference between axial-shift and the shift operation in Swin Transformer is that axial-shift adopts a grouped parallel design. Axial-shift is divided into V-Shift and H-Shift operations, which act on different attention heads and correspond to V-Rwin and H-Rwin. However, the shift operation in Swin Transformer performs the same shift operation in all heads.

Based on our proposed axis-shift operation, Rwin can realize more window interaction, thereby expanding the receptive field and improving model performance. We can find that the performance of Rwin with axial-shift is much better than the square window with shift operation in Swin Transformer from the ablation study Table 1a in the main paper.

Furthermore, the shift operation in Swin Transformer can be viewed as a special case of our axial-shift. When the axial-shift displacement distances are the same in all heads, the shift operation in each attention head is the same. Then axial-shift degenerates into the shift operation in Swin Transformer. In general, our axial-shift is more general and efficient.

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

| Method | Scale | Set5 | | Set14 | | B100 | | Urban100 | | Manga109 | |
|---|---|---|---|---|---|---|---|---|---|---|---|
| | | PSNR | SSIM | PSNR | SSIM | PSNR | SSIM | PSNR | SSIM | PSNR | SSIM |
| EDSR [14] | ×2 | 38.11 | 0.9602 | 33.92 | 0.9195 | 32.32 | 0.9013 | 32.93 | 0.9351 | 39.10 | 0.9773 |
| D-DBPN [9] | ×2 | 38.09 | 0.9600 | 33.85 | 0.9190 | 32.27 | 0.9000 | 32.55 | 0.9324 | 38.89 | 0.9775 |
| SRMDNF [28] | ×2 | 37.79 | 0.9601 | 33.32 | 0.9159 | 32.05 | 0.8985 | 31.33 | 0.9204 | 38.07 | 0.9761 |
| RDN [31] | ×2 | 38.24 | 0.9614 | 34.01 | 0.9212 | 32.34 | 0.9017 | 32.89 | 0.9353 | 39.18 | 0.9780 |
| OISR [10] | ×2 | 38.21 | 0.9612 | 33.94 | 0.9206 | 32.36 | 0.9019 | 33.03 | 0.9365 | - | - |
| RCAN [29] | ×2 | 38.27 | 0.9614 | 34.12 | 0.9216 | 32.41 | 0.9027 | 33.34 | 0.9384 | 39.44 | 0.9786 |
| NLRN [15] | ×2 | 38.00 | 0.9603 | 33.46 | 0.9159 | 32.19 | 0.8992 | 31.81 | 0.9249 | - | - |
| RNAN [30] | ×2 | 38.17 | 0.9611 | 33.87 | 0.9207 | 32.31 | 0.9014 | 32.73 | 0.9340 | 39.23 | 0.9785 |
| SRFBN [12] | ×2 | 38.11 | 0.9609 | 33.82 | 0.9196 | 32.29 | 0.9010 | 32.62 | 0.9328 | 39.08 | 0.9779 |
| SAN [4] | ×2 | 38.31 | 0.9620 | 34.07 | 0.9213 | 32.42 | 0.9028 | 33.10 | 0.9370 | 39.32 | 0.9792 |
| RFANet [16] | ×2 | 38.26 | 0.9615 | 34.16 | 0.9220 | 32.41 | 0.9026 | 33.33 | 0.9389 | 39.44 | 0.9783 |
| NSR [6] | ×2 | 38.23 | 0.9614 | 33.94 | 0.9203 | 32.34 | 0.9020 | 33.02 | 0.9367 | 39.31 | 0.9782 |
| IGNN [33] | ×2 | 38.24 | 0.9613 | 34.07 | 0.9217 | 32.41 | 0.9025 | 33.23 | 0.9383 | 39.35 | 0.9786 |
| HAN [24] | ×2 | 38.27 | 0.9614 | 34.16 | 0.9217 | 32.41 | 0.9027 | 33.35 | 0.9385 | 39.46 | 0.9785 |
| CSNLN [23] | ×2 | 38.28 | 0.9616 | 34.12 | 0.9223 | 32.40 | 0.9024 | 33.25 | 0.9386 | 39.37 | 0.9785 |
| NLSA [22] | ×2 | 38.34 | 0.9618 | 34.08 | 0.9231 | 32.43 | 0.9027 | 33.42 | 0.9394 | 39.59 | 0.9789 |
| CRAN [32] | ×2 | 38.31 | 0.9617 | 34.22 | 0.9232 | 32.44 | 0.9029 | 33.43 | 0.9394 | 39.75 | 0.9793 |
| DFSA+ [19] | ×2 | 38.38 | 0.9620 | 34.33 | 0.9232 | 32.50 | 0.9036 | 33.66 | 0.9412 | 39.98 | 0.9798 |
| IPT [3] | ×2 | 38.37 | - | 34.43 | - | 32.48 | - | 33.76 | - | - | - |
| SwinIR [13] | ×2 | 38.42 | 0.9623 | 34.46 | 0.9250 | 32.53 | 0.9041 | 33.81 | 0.9427 | 39.92 | 0.9797 |
| CAT-R (ours) | ×2 | 38.48 | 0.9625 | 34.53 | 0.9251 | 32.56 | 0.9045 | 34.08 | 0.9443 | 40.09 | 0.9804 |
| CAT-A (ours) | ×2 | 38.51 | 0.9626 | 34.78 | 0.9265 | 32.59 | 0.9047 | 34.26 | 0.9440 | 40.10 | 0.9805 |
| CAT-R+ (ours) | ×2 | 38.52 | 0.9627 | 34.59 | 0.9257 | 32.58 | 0.9047 | 34.19 | 0.9450 | 40.18 | 0.9805 |
| CAT-A+ (ours) | ×2 | 38.55 | 0.9628 | 34.81 | 0.9267 | 32.60 | 0.9048 | 34.34 | 0.9445 | 40.18 | 0.9806 |
| EDSR [14] | ×3 | 34.65 | 0.9280 | 30.52 | 0.8462 | 29.25 | 0.8093 | 28.80 | 0.8653 | 34.17 | 0.9476 |
| SRMDNF [28] | ×3 | 34.12 | 0.9254 | 30.04 | 0.8382 | 28.97 | 0.8025 | 27.57 | 0.8398 | 33.00 | 0.9403 |
| RDN [31] | ×3 | 34.71 | 0.9296 | 30.57 | 0.8468 | 29.26 | 0.8093 | 28.80 | 0.8653 | 34.13 | 0.9484 |
| OISR [10] | ×3 | 34.72 | 0.9297 | 30.57 | 0.8470 | 29.29 | 0.8103 | 28.95 | 0.8680 | - | - |
| RCAN [29] | ×3 | 34.74 | 0.9299 | 30.65 | 0.8482 | 29.32 | 0.8111 | 29.09 | 0.8702 | 34.44 | 0.9499 |
| NLRN [15] | ×3 | 34.27 | 0.9266 | 30.16 | 0.8374 | 29.06 | 0.8026 | 27.93 | 0.8453 | - | - |
| RNAN [30] | ×3 | 34.66 | 0.9290 | 30.53 | 0.8463 | 29.26 | 0.8090 | 28.75 | 0.8646 | 34.25 | 0.9483 |
| SRFBN [12] | ×3 | 34.70 | 0.9292 | 30.51 | 0.8461 | 29.24 | 0.8084 | 28.73 | 0.8641 | 34.18 | 0.9481 |
| SAN [4] | ×3 | 34.75 | 0.9300 | 30.59 | 0.8476 | 29.33 | 0.8112 | 28.93 | 0.8671 | 34.30 | 0.9494 |
| RFANet [16] | ×3 | 34.79 | 0.9300 | 30.67 | 0.8487 | 29.34 | 0.8115 | 29.15 | 0.8720 | 34.59 | 0.9506 |
| NSR [6] | ×3 | 34.62 | 0.9289 | 30.57 | 0.8475 | 29.26 | 0.8100 | 28.83 | 0.8663 | 34.27 | 0.9484 |
| IGNN [33] | ×3 | 34.72 | 0.9298 | 30.66 | 0.8484 | 29.31 | 0.8105 | 29.03 | 0.8696 | 34.39 | 0.9496 |
| HAN [24] | ×3 | 34.75 | 0.9299 | 30.67 | 0.8483 | 29.32 | 0.8110 | 29.10 | 0.8705 | 34.48 | 0.9500 |
| CSNLN [23] | ×3 | 34.74 | 0.9300 | 30.66 | 0.8482 | 29.33 | 0.8105 | 29.13 | 0.8712 | 34.45 | 0.9502 |
| NLSA [22] | ×3 | 34.85 | 0.9306 | 30.70 | 0.8485 | 29.34 | 0.8117 | 29.25 | 0.8726 | 34.57 | 0.9508 |
| CRAN [32] | ×3 | 34.80 | 0.9304 | 30.73 | 0.8498 | 29.38 | 0.8124 | 29.33 | 0.8745 | 34.84 | 0.9515 |
| DFSA+ [19] | ×3 | 34.92 | 0.9312 | 30.83 | 0.8507 | 29.42 | 0.8128 | 29.44 | 0.8761 | 35.07 | 0.9525 |
| IPT [3] | ×3 | 34.81 | - | 30.85 | - | 29.38 | - | 29.49 | - | - | - |
| SwinIR [13] | ×3 | 34.97 | 0.9318 | 30.93 | 0.8534 | 29.46 | 0.8145 | 29.75 | 0.8826 | 35.12 | 0.9537 |
| CAT-R (ours) | ×3 | 34.99 | 0.9320 | 31.00 | 0.8539 | 29.49 | 0.8154 | 29.91 | 0.8848 | 35.29 | 0.9542 |
| CAT-A (ours) | ×3 | 35.06 | 0.9326 | 31.04 | 0.8538 | 29.52 | 0.8160 | 30.12 | 0.8862 | 35.38 | 0.9546 |
| CAT-R+ (ours) | ×3 | 35.07 | 0.9324 | 31.06 | 0.8544 | 29.52 | 0.8159 | 30.05 | 0.8864 | 35.44 | 0.9548 |
| CAT-A+ (ours) | ×3 | 35.10 | 0.9327 | 31.09 | 0.8545 | 29.55 | 0.8164 | 30.21 | 0.8872 | 35.48 | 0.9550 |
| EDSR [14] | ×4 | 32.46 | 0.8968 | 28.80 | 0.7876 | 27.71 | 0.7420 | 26.64 | 0.8033 | 31.02 | 0.9148 |
| D-DBPN [9] | ×4 | 32.47 | 0.8980 | 28.82 | 0.7860 | 27.72 | 0.7400 | 26.38 | 0.7946 | 30.91 | 0.9137 |
| SRMDNF [28] | ×4 | 31.96 | 0.8925 | 28.35 | 0.7787 | 27.49 | 0.7337 | 25.68 | 0.7731 | 30.09 | 0.9024 |
| RDN [31] | ×4 | 32.47 | 0.8990 | 28.81 | 0.7871 | 27.72 | 0.7419 | 26.61 | 0.8028 | 31.00 | 0.9151 |
| OISR [10] | ×4 | 32.53 | 0.8992 | 28.86 | 0.7878 | 27.75 | 0.7428 | 26.79 | 0.8068 | - | - |
| RCAN [29] | ×4 | 32.63 | 0.9002 | 28.87 | 0.7889 | 27.77 | 0.7436 | 26.82 | 0.8087 | 31.22 | 0.9173 |
| NLRN [15] | ×3 | 31.92 | 0.8916 | 28.36 | 0.7745 | 27.48 | 0.7306 | 25.79 | 0.7729 | - | - |
| RNAN [30] | ×3 | 32.43 | 0.8977 | 28.83 | 0.7871 | 27.72 | 0.7410 | 26.61 | 0.8023 | 31.09 | 0.9149 |
| SRFBN [12] | ×4 | 32.47 | 0.8983 | 28.81 | 0.7868 | 27.72 | 0.7409 | 26.60 | 0.8015 | 31.15 | 0.9160 |
| SAN [4] | ×4 | 32.64 | 0.9003 | 28.92 | 0.7888 | 27.78 | 0.7436 | 26.79 | 0.8068 | 31.18 | 0.9169 |
| RFANet [16] | ×4 | 32.66 | 0.9004 | 28.88 | 0.7894 | 27.79 | 0.7442 | 26.92 | 0.8112 | 31.41 | 0.918 |
| NSR [6] | ×4 | 32.55 | 0.8987 | 28.79 | 0.7876 | 27.72 | 0.7414 | 26.61 | 0.8025 | 31.10 | 0.9145 |
| IGNN [33] | ×4 | 32.57 | 0.8998 | 28.85 | 0.7891 | 27.77 | 0.7434 | 26.84 | 0.8090 | 31.28 | 0.9182 |
| HAN [24] | ×4 | 32.64 | 0.9002 | 28.90 | 0.7890 | 27.80 | 0.7442 | 26.85 | 0.8094 | 31.42 | 0.9177 |
| CSNLN [23] | ×4 | 32.68 | 0.9004 | 28.95 | 0.7888 | 27.80 | 0.7439 | 27.22 | 0.8168 | 31.43 | 0.9201 |
| NLSA [22] | ×4 | 32.59 | 0.9000 | 28.87 | 0.7891 | 27.78 | 0.7444 | 26.96 | 0.8109 | 31.27 | 0.9184 |
| CRAN [32] | ×4 | 32.72 | 0.9012 | 29.01 | 0.7918 | 27.86 | 0.7460 | 27.13 | 0.8167 | 31.75 | 0.9219 |
| DFSA+ [19] | ×4 | 32.79 | 0.9019 | 29.06 | 0.7922 | 27.87 | 0.7458 | 27.17 | 0.8163 | 31.88 | 0.9266 |
| IPT [3] | ×4 | 32.64 | - | 29.01 | - | 27.82 | - | 27.26 | - | - | - |
| SwinIR [13] | ×4 | 32.92 | 0.9044 | 29.09 | 0.7950 | 27.92 | 0.7489 | 27.45 | 0.8254 | 32.03 | 0.9260 |
| CAT-R (ours) | ×4 | 32.89 | 0.9044 | 29.13 | 0.7955 | 27.95 | 0.7500 | 27.62 | 0.8292 | 32.16 | 0.9269 |
| CAT-A (ours) | ×4 | 33.08 | 0.9052 | 29.18 | 0.7960 | 27.99 | 0.7510 | 27.89 | 0.8339 | 32.39 | 0.9285 |
| CAT-R+ (ours) | ×4 | 32.98 | 0.9049 | 29.18 | 0.7963 | 27.98 | 0.7506 | 27.73 | 0.8310 | 32.35 | 0.9280 |
| CAT-A+ (ours) | ×4 | 33.14 | 0.9059 | 29.23 | 0.7968 | 28.01 | 0.7516 | 27.99 | 0.8356 | 32.52 | 0.9293 |

Table 7: Quantitative comparison (PSNR/SSIM) with state-of-the-art methods for image SR. Best and second best results are colored with red and blue.

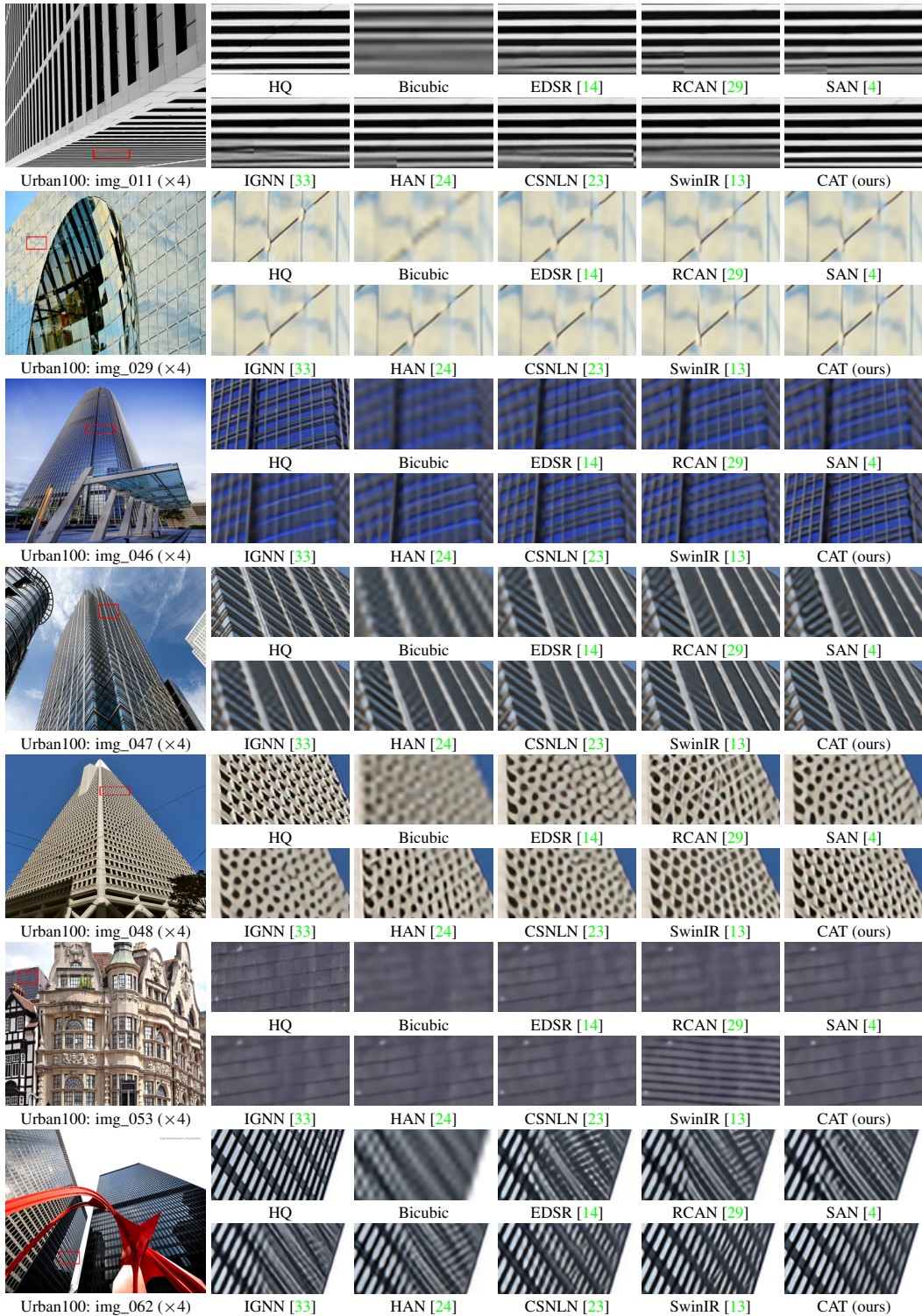

Figure 4: Visual comparison about image SR (×4) on Urban100 [11] dataset.

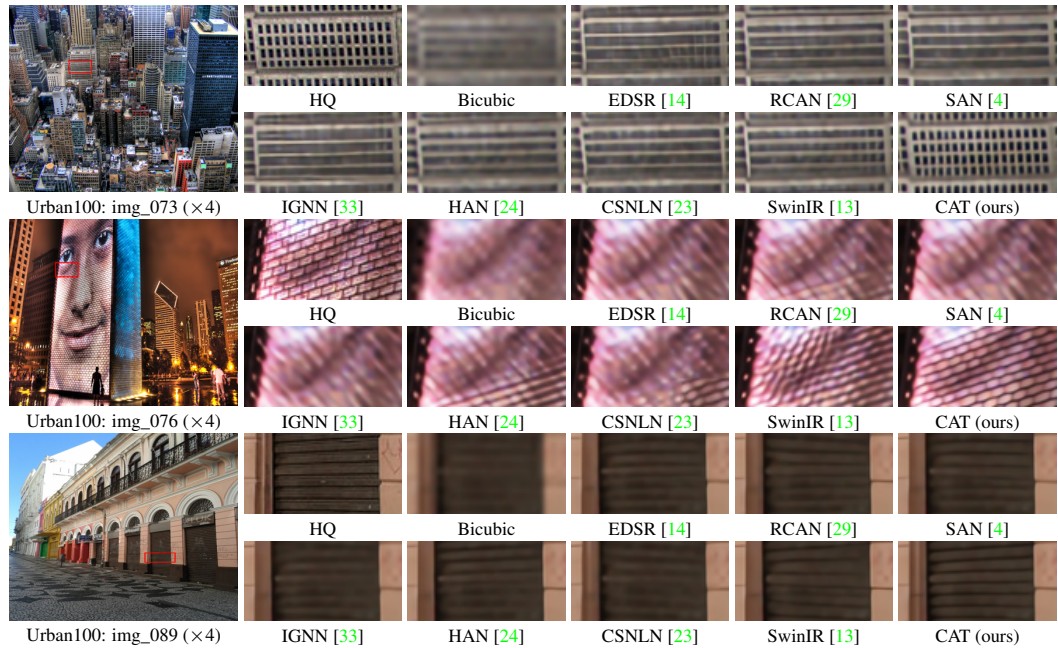

Figure 5: Visual comparison about image SR ($\times 4$) on Urban100 [11] dataset.

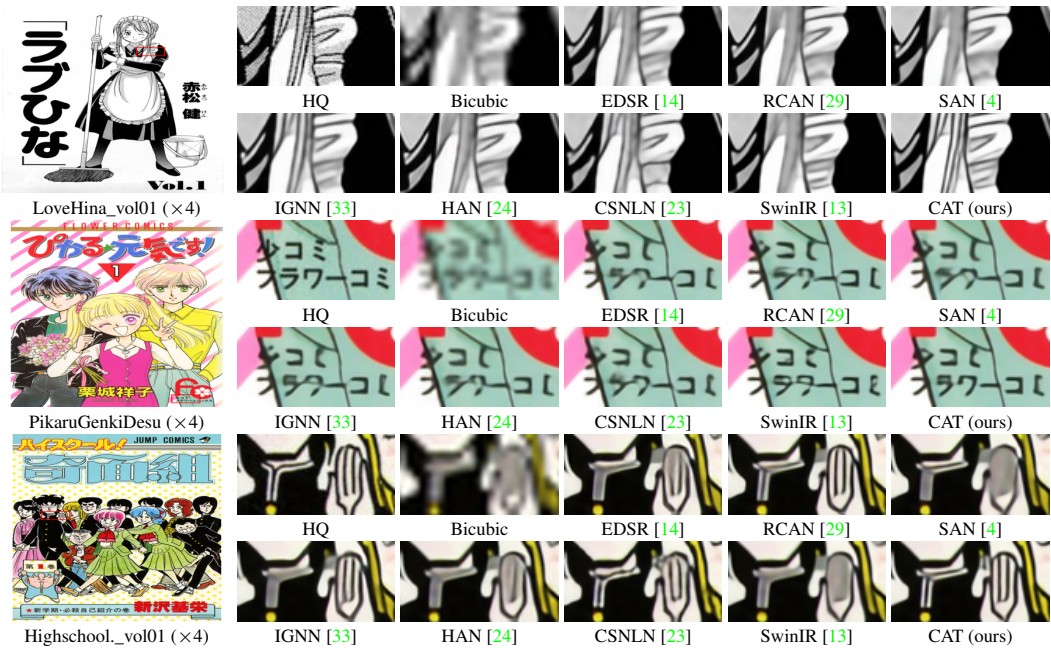

Figure 6: Visual comparison about image SR ($\times 4$) on Manga109 [21] dataset.

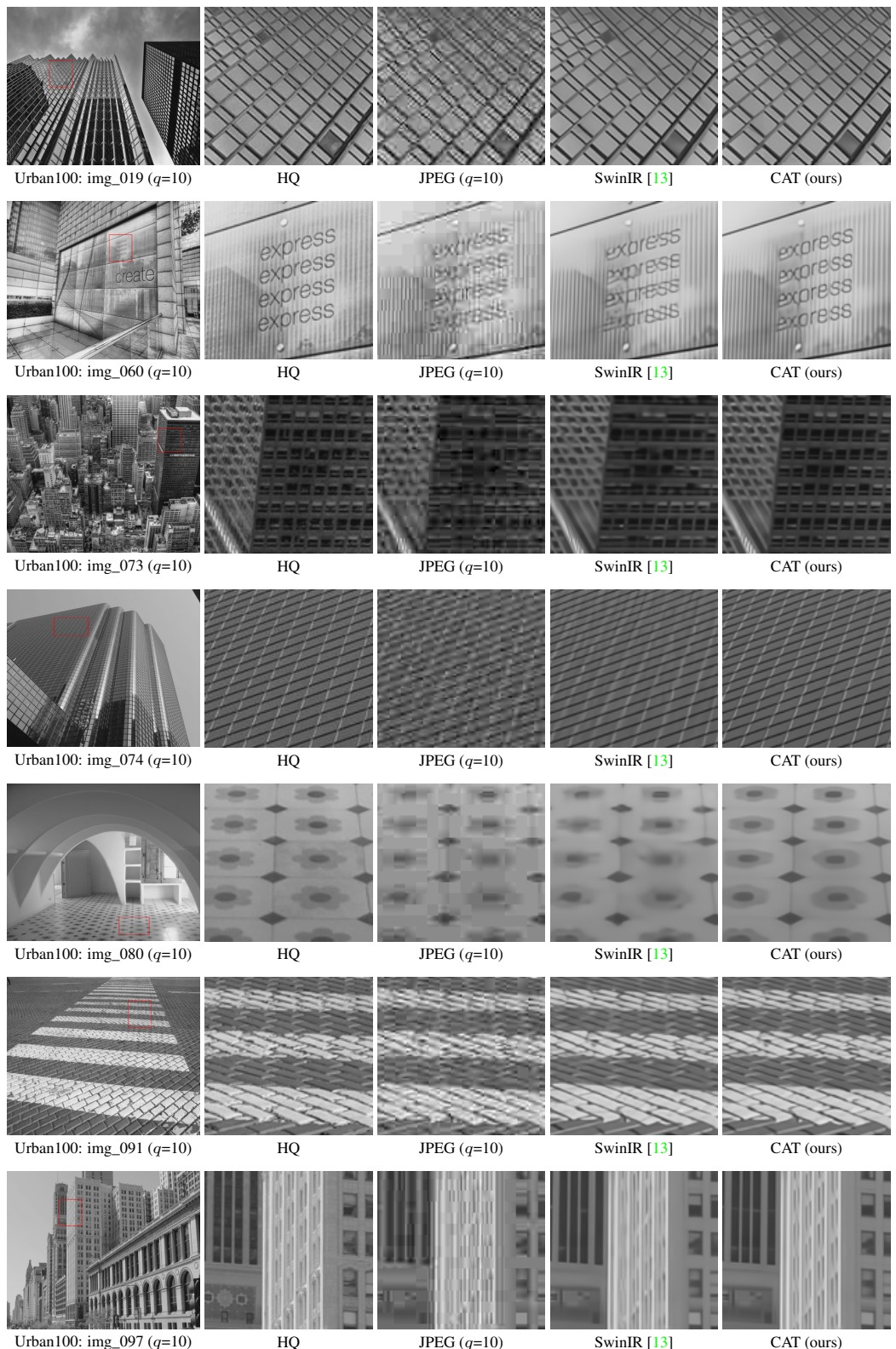

Figure 7: Visual comparison about JPEG compression artifacts reduction (*q*=10) on Urban100 [11].

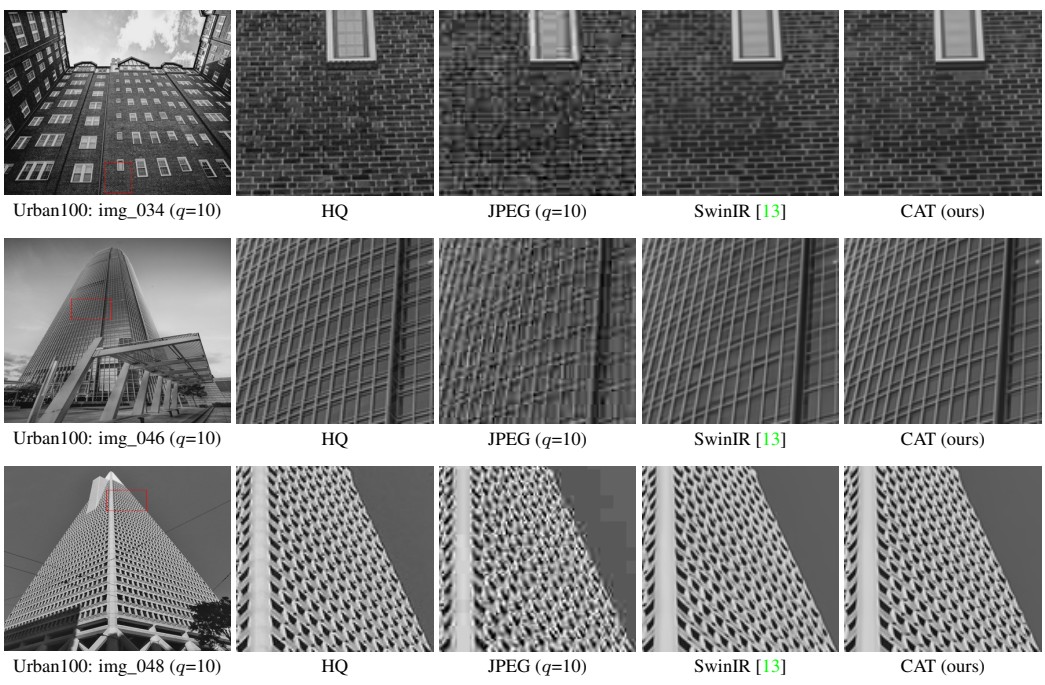

Figure 8: Visual comparison about JPEG compression artifacts reduction ($q$=10) on Urban100 [11].