# OpenReview forum: "Cross Aggregation Transformer for Image Restoration"
_NeurIPS.cc/2022/Conference — NeurIPS 2022 Accept_

### Official Review · Reviewer_J9TR · 2022-07-04

**Rating:** 8
**Confidence:** 5
**Soundness:** 4 excellent
**Presentation:** 4 excellent
**Contribution:** 4 excellent

**Summary:**

The authors propose a new image restoration model, cross aggregation Transformer (CAT). The key idea is the rectangle-window self-attention, which uses horizontal and vertical rectangle window attention in different heads parallelly. The axial-shift operation is further introduced to different window interactions. They further propose a locality complementary module to incorporate some useful properties of CNN into Transformer. The ablation study and main results support the contributions well.

**Questions:**

1. Please discuss the differences between the key ideas of the work (e.g., axial-Rwin) and CSWin.
2. The authors said in the ‘Conclusion’ part that the proposed CAT could also be used for other image restoration tasks. Did the authors try those tasks and/or get some preliminary results?


**Limitations:**

No, the authors did not discuss too much about the limitations or potential negative societal impact. It would be better if the authors can discuss some limitations.


**Strengths And Weaknesses:**

Strengths
1. The proposed method cross aggregation Transformer (CAT) is simple yet efficient. It has been a good Transformer based method for high-quality image restoration applications.
2. All proposed components are stated clearly and logically, like rectangle-window self-attention, axial-shift operation, and locality complementary module.
3. The ablation study results are extensive and demonstrate the effects of each proposed component.
4. The main comparisons with recent leading methods (e.g., SwinIR) further show the superior of the proposed one. According to Table 2, we can see the best and second-best results are almost achieved by the proposed method CAT. The visual differences between the proposed CAT and other methods are very obvious, and further show the effectiveness of the proposed CAT. Similar observations happen for image JPEG compression artifact reduction.
5. The overall writing and organization are pretty good.
6. The authors also provide code and pre-trained models for reproduction, which further shows the solidness of the work.

Weaknesses
1. In Figure 3, the meanings of H, W, sl should be given in the caption.
2. In Table 4, the authors provide model size, FLOPs, and PSNR comparisons. The proposed CAT obtains the highest performance. However, the proposed CAT has a larger model size and FLOPs than the related work SwinIR. It would be much better if the authors provide comparisons with SwinIR using similar parameter number and FLOPs.
3. Some details should be made more clear. For example, for image SR, the authors provide two versions of CAT: CAT-R and CAT-A. For JPEG compression artifact reduction, the authors only show CAT. It is not very clear about its model size and FLOPs.

---

> ### Author Response · Authors · 2022-08-02
> **Response to Reviewer J9TR (denoted as R4) part 1**
>
> `Q4-1:` In Figure 3, the meanings of H, W, sl should be given in the caption.
>
> `A4-1:` Thanks for the suggestion. We have modified the caption of Figure 3 in the paper as follows.
>
> Illustration of attention expansion, directional features aggregation, and axial-Rwin. $H$, $W$, and $sl$ mean height, width, and the other side length.
>
>
>
> `Q4-2:` In Table 4, the authors provide model size, FLOPs, and PSNR comparisons. The proposed CAT obtains the highest performance. However, the proposed CAT has a larger model size and FLOPs than the related work SwinIR. It would be much better if the authors provide comparisons with SwinIR using similar parameter number and FLOPs.
>
> `A4-2:` Thanks for the suggestion. We **already provided** a comparison of similar model sizes and FLOPs in **supplementary material** Table 2. We also show them here. Output size is 3$\times$512$\times$512 to calculate FLOPs.
>
> | Method  | Params (M) | FLOPs (G) |   Set5    |   Set14   |   B100    | Urban100  | Manga109  |
> | ------- | :--------: | :-------: | :-------: | :-------: | :-------: | :-------: | :-------: |
> | SwinIR  |   11.90    |   215.3   | **32.92** |   29.09   |   27.92   |   27.45   |   32.03   |
> | CAT-R-2 |   11.93    |   216.3   |   32.91   | **29.13** | **27.93** | **27.59** | **32.16** |
>
> Our CAT-R-2 obtains a similar model size, FLOPs with SwinIR but obtains better performance.
>
>
>
> `Q4-3:` Some details should be made more clear. For example, for image SR, the authors provide two versions of CAT: CAT-R and CAT-A. For JPEG compression artifact reduction, the authors only show CAT. It is not very clear about its model size and FLOPs.
>
> `A4-3:` Thanks for pointing it out. For JPEG compression artifact reduction, we employ **CAT-A**, and since only one model is proposed, we abbreviate it as CAT. We provide its model size and FLOPs as follows. Output size is 1$\times$128$\times$128 to calculate FLOPs, and PSNR values are tested $q$=40.
>
> | Method     | Params (M) | FLOPs (G) |   LIVE1   | Classic5  | Urban100  |
> | ---------- | :--------: | :-------: | :-------: | :-------: | :-------: |
> | SwinIR     |   11.49M   |   213.6   |   34.67   |   34.52   |   35.50   |
> | CAT (ours) |   16.20M   |   346.4   | **34.72** | **34.58** | **35.73** |
>
> Our proposed CAT outperforms SwinIR and has a larger model size and FLOPs than SwinIR. Due to time issues, we previously did not train a CAT of similar size to the SwinIR model. But we provide a model with similar parameter sizes and FLOPs to SwinIR on image SR, and our model outperforms SwinIR. Please refer to `A4-2` for more details. Moreover, we will propose a variant of CAT with a similar model size and FLOPs for JPEG compression artifact reduction in the future.
>
>
>
> `Q4-4:` Please discuss the differences between the key ideas of the work (e.g., axial-Rwin) and CSWin.
>
> `A4-4:` The main differences can be summarized into three points:
>
> **(1)** Axial-Rwin is inspired by CSwin and is close to CSwin in form, but it is only a special case of our proposed rectangle window self-attention. Regular-Rwin we proposed is a more general form and more flexible. In other words, the attention mechanism in CSwin is a special case proposed in our paper.
>
> **(2)** Some methods in high-level tasks, like image classification, use operations similar to axial-Rwin. However, in low-level tasks, such as image restoration, there is still a lack of corresponding attempts. We successfully apply axial-Rwin to image restoration.
>
> **(3)** We also propose two operations: axial-shift and LCM module, to enhance Rwin-SA.
>
> To demonstrate **the effectiveness of our CAT**, we compare the performance of CAT-A and CSwin on image SR ($\times$2). The CSwin model uses our CAT architecture and replaces our Cross Aggregation Transformer Block with the CSWin Transformer Block. The implementation details and training settings are the same  for CAT and CSwin. The FLOPs are measured when the output size is set to 3$\times$256$\times$256. The comparison results are shown in the following table.
>
> | Method | Parameters | FLOPs |   Set5    |   Set14   | B100      | Urban100  | Manga109  |
> | ------ | :--------: | :---: | :-------: | :-------: | --------- | :-------: | :-------: |
> | CSwin  |   16.45M   | 350.7 |   38.40   |   34.42   | 32.46     |   33.73   |   39.83   |
> | CAT-A  |   16.46M   | 350.7 | **38.51** | **34.78** | **32.59** | **34.26** | **40.10** |
>
> We can see from the above table that CAT-A performs much better than CSwin with a similar model size and computational complexity.

---

> ### Author Response · Authors · 2022-08-02
> **Response to Reviewer J9TR (denoted as R4) part 2**
>
> `Q4-5:` The authors said in the ‘Conclusion’ part that the proposed CAT could also be used for other image restoration tasks. Did the authors try those tasks and/or get some preliminary results?
>
> `A4-5:`  We try to apply our CAT on real image denoising and achieve some preliminary results.
>
> We apply our proposed **cross aggregation Transformer block** to the encoder-decoder-based **UNet architecture** to construct CAT (refer to Restormer). We apply our CAT to real image denoising.
>
> We use SIDD training dataset to train our CAT. The training settings and training process are the same as Restormer. We test our model on SIDD and DND datasets. The comparison of our CAT with the state-of-the-art methods: MIRNet, Uformer, and Restormer is shown in the following table.
>
> | Method                                                       | Provenance | Params |   SIDD    |           |    DND    |           |
> | ------------------------------------------------------------ | ---------- | :----: | :-------: | :-------: | :-------: | :-------: |
> |                                                              |            |        |   PSNR    |   SSIM    |   PSNR    |   SSIM    |
> | MIRNet                                                       | ECCV 2020  | 31.79M |   39.72   |   0.959   |   39.88   |   0.956   |
> | Uformer-S [used for comparison in Table 6 of Restormer]      | CVPR 2022  | 20.63M |   39.77   |   0.959   |   39.96   |   0.956   |
> | Uformer-B [results shown in CVPR 2022 camera ready]          | CVPR 2022  | 50.88M |   39.89   |   0.960   |   40.04   |   0.956   |
> | Restormer [training iterations: 300K]                        | CVPR 2022  | 26.11M | **40.02** | **0.960** |   40.03   |   0.956   |
> | CAT [ours, **completed trained**; training iterations: 300K] | our method | 25.77M |   40.01   | **0.960** | **40.05** | **0.956** |
> | CAT+ [ours, **self ensemble**]                               | our method | 25.77M | **40.05** | **0.960** | **40.08** | **0.956** |
>
>
>
> We can observe that our CAT outperforms MIRNet, Uformer-S, and Uformer-B on all datasets. And our CAT performs better than Restormer on DND and is comparable to Restormer on SIDD. Moreover, the parameter of our CAT is smaller than all models compared, except Uformer-S. Overall, our CAT outperforms or is comparable to other methods.
>
> We have added this part to the supplementary material. Please refer to **Sec. 4.1** in the **supplementary material** for details. Moreover, we have added test code and pretrained model to our source code (Anonymous Github) for reviewers to test and track.
>
> We also plan to apply our CAT to more image restoration tasks in the future.

---

> ### Author Response · Authors · 2022-08-05
> **Follow-up discussions with Reviewer R4 (J9TR)**
>
> Dear Reviewer J9TR ,
>
> Thanks for your valuable time and comments. We have responded to the related questions.
>
>
>
> **(1)** We modify the caption of **Fig. 3** in the main paper.
>
> **(2)** We provide **a variant CAT** of similar model size and FLOPs in supplementary material and merge the result into **Table 4** in the main paper.
>
> **(3)** We show **the model size and FLOPs** of CAT for JPEG compression artifact reduction.
>
> **(4)** We provide analysis and experiments to show the differences between the key ideas of our work (e.g., axial-Rwin) and **CSwin**.
>
> **(5)** Finally, we apply our method to **real Image Denoising**. The performance of our CAT is **comparable to or better** than other methods (**MIRNet, Uformer, and Restormer**).
>
>
>
> Please let us know if you have any unsolved or other concerns. Then, we have enough time to provide further feedback. Thanks.
>
> Best,
>
> Authors

---

> ### Comment · Reviewer_J9TR · 2022-08-08
> **After rebuttal**
>
> Thanks for the rebuttal. The authors provide the results of CAT applied for real image denoising, where the proposed CAT still performs comparable to or better than recent methods, like CVPR2022 papers: Uformer and Restormer. Other responses (e.g., fair comparisons with similar model size, discussions with CSWin). I have also checked the updated paper and found that the corresponding parts have been revised well. I am satisfied with the rebuttal, which has addressed all my concerns. I keep my original rating as strong accept.

---

### Official Review · Reviewer_LKyp · 2022-07-06

**Rating:** 8
**Confidence:** 5
**Soundness:** 4 excellent
**Presentation:** 4 excellent
**Contribution:** 4 excellent

**Summary:**

The authors proposed cross aggregation Transformer (CAT) for image restoration, where the long-range dependencies are well considered. The proposed CAT uses the rectangle window self-attention (Rwin-SA) with axial-shift and aggregates the features cross different windows. Plus, they proposed a locality complementary module (LCM) to capture both local and global information. The main results on classic image restoration applications, like image SR and JPEG artifact reduction, show the superior performance of CAT quantitatively and visually.

**Questions:**

(1) There are image restoration (i.e., image SR and JPEG artifact reduction) applications are conducted. The pipeline in Figure 1 mainly takes image SR as an example. Can the authors give more details about the pipeline for JPEG artifact reduction? Or the only difference is that the upscaling module is removed? Please clarify this.

(2) Is the axial-shift in CAT exactly the same as the shift operation in SwinIR? If not, please tell the differences.

(3) Can the method be applied to other image restoration tasks well? Like image denoising and deblurring. If so, how is the performance of CAT? Will CAT still achieves SOTA performance?

**Limitations:**

In supplementary file, the authors have discussed the limitations and potential negative societal impact. Overall, I agree with the views of the authors.

**Strengths And Weaknesses:**

Strengths:

(1) Firstly, the paper writing, organization, method, illustration figures (e.g., the pipeline in Figure 1), results are pretty good, which impress me a lot! I believe this paper is carefully prepared.

(2) The idea of CAT is novel and a promising step to investigate more powerful Transformer based methods for high-quality image restoration.

(3) The authors proposed cross aggregation Transformer block as the basic unit to form the image restoration networks. In the block, there three key parts: rectangle-window self-attention, axial-shift operation, and locality complementary module. The effect of each part has been demonstrated with ablation study experiments.

(4) The main results with other leading methods are also extensive. The authors conducted experiments on image SR and JPEG artifact reduction. According to the quantitative (i.e., Tables 2 and 3) and visual (i.e., Figures 5 and 6) comparisons, we can further see the superior performance of the CAT.

(5) The authors provided more variant models of the proposed one and achieved further improvements, also indicating the promising representation ability of CAT.

(6) In supplementary file, the authors provided convergence analyses on CAT-A-CAT-R, CAR-R-2, and SwinIR using multiple testing sets. These results show that various CAT models performs better than the basic SwinIR.

Weaknesses:

(1) For the model size analyses in Table 4, CAT-R and CAT-A perform better than other compared ones, but their parameter numbers are not the smallest or still higher than SwinIR. In supplementary file, the authors gave a more reasonable comparison (see Table 2), which should be included in the main paper.

(2) For JPEG artifact reduction, the quantitative results do not show too much improvement of CAT over SwinIR. It would be better if the authors can give more analyses.

---

> ### Author Response · Authors · 2022-08-02
> **Response to Reviewer LKyp (denoted as R3) part 1**
>
> `Q3-1:` For the model size analyses in Table 4, CAT-R and CAT-A perform better than other compared ones, but their parameter numbers are not the smallest or still higher than SwinIR. In supplementary file, the authors gave a more reasonable comparison (see Table 2), which should be included in the main paper.
>
> `A3-1:` Thanks for the valuable suggestion. We have merged the results (CAT-R-2) into Table 4. We also show them here. Output size is 3$\times$512$\times$512 to calculate FLOPs, and PSNR values are tested on Urban100 ($\times$4).
>
> | Method         | EDSR  | RCAN  |  HAN  |  CSNLN   | SwinIR | CAT-R (ours) | CAT-A (ours) | CAT-R-2 (ours) |
> | -------------- | :---: | :---: | :---: | :------: | :----: | :----------: | :----------: | :------------: |
> | PSNR (dB)      | 26.64 | 26.82 | 26.85 |  27.22   | 27.45  |    27.62     |    27.89     |     27.59      |
> | FLOPs (G)      | 823.3 | 261.0 | 269.1 | 84,155.2 | 215,3  |    292.7     |    360.7     |     216.3      |
> | Parameters (M) | 43.09 | 15.59 | 16.07 |   6.57   | 11.90  |    16.60     |    16.60     |     11.93      |
>
>
>
> `Q3-2:` For JPEG artifact reduction, the quantitative results do not show too much improvement of CAT over SwinIR. It would be better if the authors can give more analyses.
>
> `A3-2:` Thanks for the suggestion. We do more analysis on this part as follows.
>
> **(1)** Our CAT has a more robust representational ability to recover structural contents and texture details due to rectangle window self-attention. However, for the JPEG artifact reduction testing datasets: LIVE1 and Classic5, the number of images they contain is small (5 and 29), and the texture features are not rich. So the overall improvement effect is not obvious. Nevertheless, the difference is evident from the visual comparison (like Fig. 6 carnivaldolls).
>
> **(2)** In the **supplementary material** Table 3, we test CAT and SwinIR on the Urban100 dataset, which with more directional and repetitive texture features. The performance of CAT is much better than that of SwinIR, with a **0.25 dB** improvement. We also show them here (only PSNR).
>
> | Method |   q=10    |   q=20    |   q=30    |   q=40    |
> | ------ | :-------: | :-------: | :-------: | :-------: |
> | SwinIR |   30.55   |   33.12   |   34.58   |   35.50   |
> | CAT    | **30.80** | **33.38** | **34.81** | **35.73** |
>
>
>
> `Q3-3:` There are image restoration (i.e., image SR and JPEG artifact reduction) applications are conducted. The pipeline in Figure 1 mainly takes image SR as an example. Can the authors give more details about the pipeline for JPEG artifact reduction? Or the only difference is that the upscaling module is removed? Please clarify this.
>
> `A3-3:` The only difference between the pipeline for JPEG artifact reduction and image SR is the reconstruction module. Other modules: shallow feature extraction and deep feature extraction, remain unchanged.
>
> **For JPEG artifact reduction**, the reconstruction is just a convolution layer. And the low-quality input $I_{LQ}$ is then added to the convolution output to produce a high-quality output $I_{HQ}$. **For image SR**, the reconstruction module is an upsampling and two convolution layers.
>
> The pseudo-code for **the reconstruction module** is as follows.
>
> ```python
> # input is the low-quality input
> # _x is the ouput of the deep feature extraction module
>
> # for image SR
> x = convolution(_x)
> output = convolution((upsample(x))
> # for JPEG compression artifact reduction
> output = input + convolution(_x)
> ```

---

> ### Author Response · Authors · 2022-08-02
> **Response to Reviewer LKyp (denoted as R3) part 2**
>
>
>
> `Q3-4:` Is the axial-shift in CAT exactly the same as the shift operation in SwinIR? If not, please tell the differences.
>
> `A3-4:` The axial-shift is different from the shift operation in SwinIR. We clarify them as follows.
>
> **(1)** Our axial-shift references the design of the shift option in Swin Transformer. However, our axial-shift adopts a grouped parallel design. Axial-shift is divided into V-Shift and H-Shift operations, which act on different attention heads and correspond to V-Rwin and H-Rwin. However, the shift operation in Swin Transformer performs the same shift operation in all heads.
>
> **(2)** Based on axial-shift, Rwin can achieve more window interactions, thereby expanding the receptive field and improving model performance. We can find that the performance of Rwin with axial-shift is much better than the square window with shift operation in Swin Transformer from **ablation study** Table 1 (a). We show them here. Output size is 3$\times$256$\times$256 to calculate FLOPs, and PSNR values are tested on Urban100 ($\times$2).
>
> | Network                   |   PSNR    |    SSIM    | FLOPs  |
> | ------------------------- | :-------: | :--------: | :----: |
> | Square. w/ shift          |   32.75   |   0.9347   | 281.8G |
> | Rectangle. w/ axial-shift | **32.91** | **0.9360** | 281.8G |
>
> **(3)** Furthermore, the shift operation in Swin Transformer can be viewed as a special case of our axial-shift. When the axial-shift displacement distances are the same in all heads, the shift operation in each attention head is the same. Then axial-shift degenerates into the shift operation in Swin Transformer. Overall, our axial-shift is more general and efficient.
>
>
>
> `Q3-5:` Can the method be applied to other image restoration tasks well? Like image denoising and deblurring. If so, how is the performance of CAT? Will CAT still achieves SOTA performance?
>
> `A3-5:` Our method can be applied to other image restoration tasks (like real image denoising) and outperforms or is comparable to other methods.
>
> We apply our proposed **cross aggregation Transformer block** to the encoder-decoder-based **UNet architecture** to construct CAT (refer to Restormer). We apply our CAT to real image denoising.
>
> We use SIDD training dataset to train our CAT. The training settings and training process are the same as Restormer. We test our model on SIDD and DND datasets. The comparison of our CAT with the state-of-the-art methods: MIRNet, Uformer, and Restormer is shown in the following table.
>
> | Method                                                       | Provenance | Params |   SIDD    |           |    DND    |           |
> | ------------------------------------------------------------ | ---------- | :----: | :-------: | :-------: | :-------: | :-------: |
> |                                                              |            |        |   PSNR    |   SSIM    |   PSNR    |   SSIM    |
> | MIRNet                                                       | ECCV 2020  | 31.79M |   39.72   |   0.959   |   39.88   |   0.956   |
> | Uformer-S [used for comparison in Table 6 of Restormer]      | CVPR 2022  | 20.63M |   39.77   |   0.959   |   39.96   |   0.956   |
> | Uformer-B [results shown in CVPR 2022 camera ready]          | CVPR 2022  | 50.88M |   39.89   |   0.960   |   40.04   |   0.956   |
> | Restormer [training iterations: 300K]                        | CVPR 2022  | 26.11M | **40.02** | **0.960** |   40.03   |   0.956   |
> | CAT [ours, **completed trained**; training iterations: 300K] | our method | 25.77M |   40.01   | **0.960** | **40.05** | **0.956** |
> | CAT+ [ours, **self ensemble**]                               | our method | 25.77M | **40.05** | **0.960** | **40.08** | **0.956** |
>
>
>
> We can observe that our CAT outperforms MIRNet, Uformer-S, and Uformer-B on all datasets. And our CAT performs better than Restormer on DND and is comparable to Restormer on SIDD. Moreover, the parameter of our CAT is smaller than all models compared, except Uformer-S. Overall, our CAT outperforms or is comparable to other methods.
>
> We have added this part to the supplementary material. Please refer to **Sec. 4.1** in the **supplementary material** for details. Moreover, we have updated the test code and pretrained model to our source code (in **Anonymous Github**) for reviewers to test and check.
>
> We also plan to apply our CAT to more image restoration tasks in the future.

---

> ### Author Response · Authors · 2022-08-05
> **Follow-up discussions with Reviewer R3 (LKyp)**
>
> Dear Reviewer LKyp ,
>
> Thanks for your valuable time and comments. We have responded to the related questions.
>
>
>
> **(1)** We merge the result (CAT-R-2) into **Table 4** in the main paper.
>
> **(2)** We explain the **small improvement** in quantitative results for JPEG artifact reduction.
>
> **(3)** We clarify the **pipeline** for JPEG artifact reduction and provide the **pseudo-code** of the reconstruction module.
>
> **(4)** We clarify the differences between **axial-shift and shift operation** in Swin Transformer.
>
> **(5)** Finally,  we apply our method to **real Image Denoising**. The performance of our CAT is **comparable to or better** than other methods (**MIRNet, Uformer, and Restormer**).
>
>
>
> Please let us know if you have any unsolved or other concerns. Then, we have enough time to provide further feedback. Thanks.
>
> Best,
>
> Authors

---

> > ### Comment · Reviewer_LKyp · 2022-08-08
> > **After rebuttal**
> >
> > Thanks for providing very detailed feedback with extensive experiments and explanations. All my concerns have been well addressed. The authors provided the real image denoising comparisons with CVPR’22 papers and still achieved good performance. I notice that the authors started to adopt their method for real image denoising and updated their results/logs/models during the discussion period, which convinced me a lot!
> >
> > I also read the comments from other reviewers. The new experiments in the rebuttal make the paper more solid. So, I would like to keep my original rating and recommend acceptance.

---

### Official Review · Reviewer_RMwD · 2022-07-07

**Rating:** 4
**Confidence:** 4
**Soundness:** 2 fair
**Presentation:** 3 good
**Contribution:** 2 fair

**Summary:**

This paper proposes an image restoration model, named Cross Aggregation Transformer (CAT). Specifically, this paper designs the Rectangle-Window Self-Attention (Rwin-SA), which utilizes horizontal and vertical rectangle window multi-head attention parallelly. The proposed Locality Complementary Module incorporates the inductive bias of CNN into Transformer to complement the self-attention mechanism.  Ablation experiments prove that the LCM improves model performance with negligible computational cost. A series experiments on different datasets demonstrate that the CAT outperforms recent state-of-art methods on two image restoration tasks: image SR and JPEG compression artifacts reduction.


**Questions:**

1. In Swin Transformer [19], Hu et al. proposed a shifted window operation, whose shifting directions include horizontal and vertical, and the shifting range of each direction is [0, k], where k is the size of the window. In this paper, author proposes a axial-shift (the horizontal one is called H-shift and the vertical one is called V-shift) operation for Rwin-SA, whose shifting range is set as sh/2 and sw/2, where sh and sw are the size of the rectangle window. The difference between shifted window operation [19] and axial-shift operation is caused by the shape of the window, so I personally think axial-shift and shifted window operation [19] are the same. Please explain in more detail the difference between V-shift and H-shift, and the difference between axial-shift and shifted window operation [19].

2.this paper proposes a Rectangle-Window Self-Attention (Rwin-SA), which utilizes horizontal and vertical rectangle window (H-Rwin and V-Rwin) attention in different heads parallelly. Furthermore, authors also proposes an axial rectangle window (axis-Rwin), in which the length of one side is fixed as the image resolution H or W. However, the axial rectangle window division method has been proposed in CSWin [10], so this part should be marked in reference [10].

3. More recent methods should be compared to demonstrate the superiority of the proposed method, such as Uformer and Restormer.
4. In related work, the discussion of the existing tranformer methods is insufficient, and the author should analyze CSWin and Uformer.
5. More experiments on color JPEG image restoration should be conducted.

**Limitations:**

The contribution and novelty of this paper are not convincing enough. More recent methods should be compared to demonstrate the superiority of the proposed method.

**Strengths And Weaknesses:**

Strengths:
1. This paper proposes an image restoration model, named Cross Aggregation Transformer (CAT). Specifically, this paper designs the Rectangle-Window Self-Attention (Rwin-SA), which utilizes horizontal and vertical rectangle window multi-head attention parallelly.
2. The proposed Locality Complementary Module incorporates the inductive bias of CNN into Transformer to complement the self-attention mechanism.  Ablation experiments prove that the LCM improves model performance with negligible computational cost.

Weakness:
1. the contribution and novelty of this paper are not convincing enough. The proposed axial rectangle window has been designed in other work. Besides, the proposed axial-shift operation is the same as shifted window operation in Swin transformer.
2. In this paper, the discussion of the existing transformer methods is insufficient.
3. More recent methods should be compared to demonstrate the superiority of the proposed method.

---

> ### Author Response · Authors · 2022-08-02
> **Response to Reviewer RMwD (denoted as R2) part 1**
>
> `Q2-1:` the contribution and novelty of this paper are not convincing enough. The proposed axial rectangle window has been designed in other work. Besides, the proposed axial-shift operation is the same as shifted window operation in Swin transformer.
>
> `A2-1:` Our work has enough contribution and novelty. We clarify them as follows.
>
> **For "axial rectangle window has been designed in other work".** Although our axial-Rwin is inspired by CSwin, the method proposed in this paper is still different from CSwin.
>
> **(1)** Axial-Rwin is inspired by CSwin and is close to CSwin in form, but it is only a special case of our proposed rectangle window self-attention. Regular-Rwin we proposed is a more general form and more flexible. In other words, the attention mechanism in CSwin is a special case proposed in our paper.
>
> **(2)** Some methods in high-level tasks, like image classification, use operations similar to axial-Rwin. However, in low-level tasks, such as image restoration, there is still a lack of corresponding attempts. We successfully apply axial-Rwin to image restoration.
>
> To demonstrate **the effectiveness of our CAT**, we compare the performance of CAT-A and CSwin on image SR ($\times$2). The CSwin model uses our CAT architecture and replaces our Cross Aggregation Transformer Block with the CSWin Transformer Block. The implementation details and training settings are the same  for CAT and CSwin. The FLOPs are measured when the output size is set to 3$\times$256$\times$256. The comparison results are shown in the following table.
>
> | Method | Parameters | FLOPs |   Set5    |   Set14   | B100      | Urban100  | Manga109  |
> | ------ | :--------: | :---: | :-------: | :-------: | --------- | :-------: | :-------: |
> | CSwin  |   16.45M   | 350.7 |   38.40   |   34.42   | 32.46     |   33.73   |   39.83   |
> | CAT-A  |   16.46M   | 350.7 | **38.51** | **34.78** | **32.59** | **34.26** | **40.10** |
>
> We can see from the above table that CAT-A performs much better than CSwin with a similar model size and computational complexity.
>
> **For "axial-shift operation is the same as shifted window operation in Swin transformer".** Our axial-shift references the design of the shift operation in Swin Transformer. However, the axial-shift we proposed is different from the shifted window operation in Swin Transformer.
>
> **(1)** The most significant difference between axial-shift and the shift operation in Swin Transformer is that axial-shift adopts **a grouped parallel design**. Axial-shift is divided into V-Shift and H-Shift operations, which act on different attention heads and correspond to V-Rwin and H-Rwin. However, the shift operation in Swin Transformer performs the same shift operation in all heads.
>
> **(2)** Based on axis-shift, Rwin can realize more window interaction, thereby expanding the receptive field and improving model performance. We can find that the performance of Rwin with axial-shift is much better than the square window with shift operation in Swin Transformer from **ablation study** Table 1 (a). We show them here. Output size is 3$\times$256$\times$256 to calculate FLOPs, and PSNR values are tested on Urban100 ($\times$2).
>
> | Network                   |   PSNR    |    SSIM    | FLOPs  |
> | ------------------------- | :-------: | :--------: | :----: |
> | Square w/o shift          |   32.50   |   0.9325   | 281.8G |
> | Square w/ shift           |   32.75   |   0.9347   | 281.8G |
> | Rectangle w/o axial-shift |   32.66   |   0.9334   | 281.8G |
> | Rectangle w/ axial-shift  | **32.91** | **0.9360** | 281.8G |
>
> **(3)** Furthermore, the shift operation in Swin Transformer can be viewed as a special case of our axial-shift. When the axial-shift displacement distances are the same in all heads, the shift operation in each attention head is the same. Then axial-shift degenerates into the shift operation in Swin Transformer. In general, our axial-shift is more general and efficient.
>
> **Overall.** Our work has enough contribution and novelty. We propose rectangle window attention (axial-Rwin is a special case) and successfully apply it to image restoration. We also propose axial-shift that matches Rwin and locality complementary module (LCM) to enhance Rwin. Moreover, our proposed CAT performs well and drives the development of Transformer in image restoration.

---

> ### Author Response · Authors · 2022-08-02
> **Response to Reviewer RMwD (denoted as R2) part 2**
>
> `Q2-2:`  In this paper, the discussion of the existing transformer methods is insufficient.
>
> `A2-2:` Thanks for pointing it out. We have modified our paper (related work). And we discuss more Transformer methods below.
>
> **(1)** Although Transformer has made breakthroughs in the computer vision field, its quadratic complexity of input size limits the scope of use. Many works have been proposed to reduce the computational complexity. **Swin** uses local window attention and realizes the interaction between windows through shift operations. But the 8$\times$8 square window limits the receptive field of the Transformer. **PVT** reduces the size of keys ($K$) and values ($V$) through projection, but this operation may cause information loss. **HaloNet** **[ref1]** uses local window attention and realizes the interaction between windows through halo operations, but its receptive field grows slowly. **CSWin** proposes cross-shaped window attention, but simple horizontal and vertical stripes require a large window to obtain a sufficient receptive field. Therefore, it is difficult to guarantee a sufficient receptive field within the limited complexity. **Twins** uses a combination of global and local attention operations, and **XCiT** **[ref2]** uses channel-attention instead of spatial-attention.
>
> **(2)** Meanwhile, many methods propose using convolution to enhance Transformer. For example, **CPVT** uses convolution to calculate position encoding. **CMT** **[ref3]** uses convolution to replace linear projection in multi-head self-attention (MHSA). However, these methods cannot effectively add local information to Transformer.
>
> **(3)** The **Uformer** model is also proposed in image restoration, using U-Net structure and local window attention. It also adds depthwise convolution in feed-forward network (FFN). But it is limited by the lack of direct interaction between windows.
>
> **(4)** However, our proposed CAT model, through Rwin-SA, can effectively realize the interaction between windows while ensuring linear complexity. And further increase the window interaction by axial-shift. Moreover, the LCM module realizes the global and local coupling.
>
> **[ref1]** Ashish Vaswani, Prajit Ramachandran, Aravind Srinivas, Niki Parmar, Blake Hechtman, and Jonathon Shlens. Scaling local self-attention for parameter efficient visual backbones. In CVPR, 2021
>
> **[ref2]** AlaaeldinAli, HugoTouvron, MathildeCaron, PiotrBojanowski, MatthijsDouze, ArmandJoulin, Ivan Laptev, Natalia Neverova, Gabriel Synnaeve, Jakob Verbeek, et al. Xcit: Cross-covariance image transformers. In NeurIPS, 2021
>
> **[ref3]** Jianyuan Guo, Kai Han, Han Wu, Yehui Tang, Xinghao Chen, Yunhe Wang, and Chang Xu. Cmt: Convolutional neural networks meet vision transformers. In CVPR, 2022

---

> ### Author Response · Authors · 2022-08-02
> **Response to Reviewer RMwD (denoted as R2) part 3**
>
> `Q2-3:` More recent methods should be compared to demonstrate the superiority of the proposed method.
>
> `A2-3:` Thanks for the valuable suggestions for comparing our method with other methods. Following the suggestions of R2, we compare our method with more methods like Uformer and Restormer.
>
> We apply our proposed **cross aggregation Transformer block** to the encoder-decoder-based **UNet architecture** to construct CAT (refer to Restormer). We apply our CAT to real image denoising.
>
> We use SIDD training dataset to train our CAT. The training settings (**totaling 300K iterations**) and training process are the same as Restormer. We test our model on SIDD and DND datasets. The comparison of our CAT with the state-of-the-art methods: MIRNet, Uformer, and Restormer, is shown in the following table.
>
> | Method                                                       | Provenance | Params |   SIDD    |           |    DND    |           |
> | ------------------------------------------------------------ | ---------- | :----: | :-------: | :-------: | :-------: | :-------: |
> |                                                              |            |        |   PSNR    |   SSIM    |   PSNR    |   SSIM    |
> | MIRNet                                                       | ECCV 2020  | 31.79M |   39.72   |   0.959   |   39.88   |   0.956   |
> | Uformer-S [used for comparison in Table 6 of Restormer]      | CVPR 2022  | 20.63M |   39.77   |   0.959   |   39.96   |   0.956   |
> | Uformer-B [results shown in CVPR 2022 camera ready]          | CVPR 2022  | 50.88M |   39.89   |   0.960   |   40.04   |   0.956   |
> | Restormer [training iterations: 300K]                        | CVPR 2022  | 26.11M | **40.02** | **0.960** |   40.03   |   0.956   |
> | CAT [ours, **completed trained**; training iterations: 300K] | our method | 25.77M |   40.01   | **0.960** | **40.05** | **0.956** |
> | CAT+ [ours, **self ensemble**]                               | our method | 25.77M | **40.05** | **0.960** | **40.08** | **0.956** |
>
>
>
> **(1)** We can observe that our CAT outperforms MIRNet, Uformer-S, and Uformer-B on all datasets. And our CAT performs better than Restormer on DND and is comparable to Restormer on SIDD. Moreover, the parameter of our CAT is smaller than all models compared, except Uformer-S.
>
> **(2)** Meanwhile, if we further enlarge the model size of our CAT, the model performance will improve further. This conclusion can be drawn from the comparison of Uformer-S and Uformer-B.
>
> **All these results demonstrate the superiority of our proposed method.**
>
> **(3)** We have added this part to the supplementary material. Please refer to **Sec. 4.1** in the **supplementary material** for details (the online test result of DND is shown in **Fig. 3**). Moreover, we have updated the test code and pretrained model to our source code (in **Anonymous Github**) for reviewers to test and check.

---

> ### Author Response · Authors · 2022-08-02
> **Response to Reviewer RMwD (denoted as R2) part 4**
>
> `Q2-4:` In Swin Transformer [19], Hu et al. proposed a shifted window operation, whose shifting directions include horizontal and vertical, and the shifting range of each direction is [0, k], where k is the size of the window. In this paper, author proposes a axial-shift (the horizontal one is called H-shift and the vertical one is called V-shift) operation for Rwin-SA, whose shifting range is set as sh/2 and sw/2, where sh and sw are the size of the rectangle window. The difference between shifted window operation [19] and axial-shift operation is caused by the shape of the window, so I personally think axial-shift and shifted window operation [19] are the same. Please explain in more detail the difference between V-shift and H-shift, and the difference between axial-shift and shifted window operation [19].
>
> `A2-4:` Thanks for your question. We clarify them as follows.
>
> **For "the difference between V-shift and H-shift".** For V-shift and H-shift, the implementation is the same, but the shift distance is different. In a special case: the height and width of V-Rwin is [$h'$, $w'$] and H-Rwin is [$w'$, $h'$]. Here $h'$ and $w'$ are constants. Then the displacement distance of V-Shift is ($\frac{h'}{2}$, $\frac{w'}{2}$), and H-shift is ($\frac{w'}{2}$, $\frac{h'}{2}$), where the first parameter represents the downward distance, the second represents the left distance.
>
> **For "the difference between axial-shift and shifted window operation".**
>
> **(1)** Our axial-shift references the design of the shift option in Swin Transformer. However, our proposed axial-shift operation adopts a grouped parallel design. Different shift distances are used to match the corresponding Rwin-SA (V-Rwin and H-Rwin). Specifically, **we use different shift operations in different attention heads**, and the shift operation corresponds to the attention mechanism in the head.
>
> **(2)** However, the shift operation in Swin Transformer performs the same shift operation on all heads. This issue makes it hard to achieve sufficient interaction between windows, limiting the growth of the receptive field.
>
> **(3)** Furthermore, the shift operation in Swin Transformer can be viewed as a special case of our axial-shift. When the axial-shift displacement distances are the same in all heads, the shift operation in each attention head is the same. Then it degenerates into the shift operation in Swin Transformer.
>
> **We have responded to another similar question, `Q2-1`. Please refer to `A2-1` for more details.**
>
>
>
> `Q2-5:` this paper proposes a Rectangle-Window Self-Attention (Rwin-SA), which utilizes horizontal and vertical rectangle window (H-Rwin and V-Rwin) attention in different heads parallelly. Furthermore, authors also proposes an axial rectangle window (axis-Rwin), in which the length of one side is fixed as the image resolution H or W. However, the axial rectangle window division method has been proposed in CSWin [10], so this part should be marked in reference [10].
>
> `A2-5:` Thanks for pointing it out. We have marked "the axial rectangle window division method has been proposed in CSWin" in our main paper. However, the method proposed in this paper is still different from CSwin.
>
> **(1)** Axial-Rwin is inspired by CSwin and is close to CSwin in form, but it is only a special case of our proposed rectangle window self-attention. Regular-Rwin we proposed is a more general form and more flexible. In other words, the attention mechanism in CSwin is a special case proposed in our paper.
>
> **(2)** At the same time, some methods in high-level tasks, like image classification, use operations similar to axial-Rwin. However, in low-level tasks, such as image restoration, there is still a lack of corresponding attempts. We successfully apply axial-Rwin to image restoration.
>
> **We have responded to another similar question, `Q2-1`. Please refer to `A2-1` for more details.**
>
>
>
> `Q2-6:` More recent methods should be compared to demonstrate the superiority of the proposed method, such as Uformer and Restormer.
>
> `A2-6:` Thanks for the valuable suggestions. Following the suggestions of R2, we compare with more methods like Uformer and Restormer on image denoising (real image denoising). We build our CAT model with our proposed **cross aggregation Transformer block** and the encoder-decoder-based **UNet architecture** (refer to Restormer).
>
> The experiment and comparison show that our CAT outperforms or is comparable to other methods, such as Uformer and Restormer. **We have responded to another similar question, `Q2-3`. Please refer to `A2-3` for more details.**

---

> ### Author Response · Authors · 2022-08-02
> **Response to Reviewer RMwD (denoted as R2) part 5**
>
> `Q2-7:` In related work, the discussion of the existing tranformer methods is insufficient, and the author should analyze CSWin and Uformer.
>
> `A2-7:` Thanks for pointing it out. We further analyze CSWin and Uformer and add this part to related work.
>
> **CSWin** computes self-attention in the horizontal and vertical stripes in parallel. The parallel strategy can enlarge the receptive area without introducing additional computational costs. However, the width of the stripes is crucial for the receptive field. Small width limits the receptive field, thus affecting the performance of the model. However, increasing the stripe width increases the computational complexity, which is not conducive to model application.  This problem comes from the lack of interaction between the stripes, which limits the growth of the receptive field.
>
> In addition, The **Uformer** [ref4] model is also proposed in image restoration, using U-Net structure and local window attention. It also adds depthwise convolution in feed-forward network (FFN). But it is limited by the lack of direct interaction between windows.
>
> We also discuss other Transformer methods further. **CeiT** uses convolution operations in token embedding and FFN to enhance Transformer locality. **Focal** incorporates both fine-grained local and coarse-grained global interactions. For a discussion of other Transformer methods, **please refer to `A2-2`**.
>
>
>
> `Q2-8:` More experiments on color JPEG image restoration should be conducted.
>
> `A2-8:` Thanks for suggesting applying our method to color JPEG image restoration. We apply our CAT and SwinIR to **color JPEG compression artifact reduction** with JPEG compression qualities of 40. We train on DIV2K, Flickr2K, BSD500, and WED. And we have four testing datasets: Set5, Set14, LIVE1, and Urban100, with JPEG compression qualities of 40. We compare our CAT with SwinIR.
>
> Due to time issues, we only completed part of the training (fininshed iterations = 200K, target total iterations = 1600K) for both CAT and SwinIR. But under the same training conditions, our CAT outperforms SwinIR on all datasets. Quantitative comparisons are shown in the following table.
>
> | Method |   Set5    |            |   Set14   |            |   LIVE1   |            | Urban100  |            |
> | ------ | :-------: | :--------: | :-------: | :--------: | :-------: | :--------: | :-------: | :--------: |
> |        |   PSNR    |    SSIM    |   PSNR    |    SSIM    |   PSNR    |    SSIM    |   PSNR    |    SSIM    |
> | SwinIR |   37.44   |   0.9487   |   35.74   |   0.9319   |   35.01   |   0.9370   |   35.42   |   0.9520   |
> | CAT    | **37.51** | **0.9491** | **35.87** | **0.9326** | **35.11** | **0.9380** | **35.76** | **0.9536** |
>
> We have added this part to the supplementary material and provided more comparative results. Please refer to **Sec. 4.2** in the **supplementary material** for details.

---

> ### Author Response · Authors · 2022-08-05
> **Follow-up discussions with Reviewer R2 (RMwD)**
>
> Dear Reviewer RMwD ,
>
> Thanks for your valuable time and comments. We have responded to the related questions.
>
> **(1)** We explain our **contribution and novelty**, including analysis and experiments to demonstrate the differences between axial-Rwin and **CSwin**. Furthermore, we analyze the difference between axial-shift and shift operations in Swin Transformer.
>
> **(2)** We discuss more existing Transformer methods and modify the **related work** in the main paper.
>
> **(3)** We compare our method with **Uformer and Restormer** on real image denoising. The **comparable or better performance** of our CAT demonstrates the superiority of our method.
>
> **(4)** Finally, we experiment on **color JPEG compression artifact reduction**, and our method **outperforms SwinIR**.
>
> Please let us know if you have any unsolved or other concerns. Then, we have enough time to provide further feedback. Thanks.
>
> Best,
>
> Authors

---

> ### Author Response · Authors · 2022-08-08
> **Second call for follow-up discussions with Reviewer R2 (RMwD)**
>
> Dear Reviewer RMwD ,
>
>
> Thanks again for your precious time and valuable comments.
>
>
> After rebuttal, Reviewer mWKh (denoted as R1), Reviewer LKyp (denoted as R3), and Reviewer J9TR (denoted as R4)  now agree that our response solves their concerns. The concerns include several points as follows.
>
> **(1)** The differences between **axial-shift and shift operation** in Swin Transformer.
>
> **(2)** The differences between CAT (or axial-Rwin) and **CSwin**.
>
> **(3)** Apply our CAT to other image restoration tasks, and compare with more methods (e.g., **Uformer and Restormer**).
>
>
>
> Are there any deficiencies in our rebuttal? Whether the corresponding responses and results we provide cover your concerns? The discussion period ending date is **August 9**. Please let us know if you have any unsolved or other concerns. Thanks.
>
>
>
> Best,
>
> Authors

---

> ### Author Response · Authors · 2022-08-09
> **Third call for follow-up discussions with Reviewer R2 (RMwD) - deadline is Aug. 9**
>
> Dear Reviewer **RMwD**,
>
> In the discussion period, **all other three reviewers** have discussed with us and admitted that we have **addressed their concerns** well.
>
> We have sent you **several calls for discussions**. But, you still have not responded to us.
>
> We believe it is **professional** to participate in the discussion before the ending date **August 9**.
>
> **Could you please provide your feedback?** Thanks.
>
> Best,
>
> Authors

---

> > ### Comment · Reviewer_RMwD · 2022-08-09
> > **After rebuttal**
> >
> > Sorry for the late. But I have to say that the openreivew system is very unfriendly for reviewers. I spent lots of time to find a way to access my comments and make a response.
> > Thanks for author's response. I see that authors provided multiple additional experiments to support their contribution.
> > However, I still concern about the novelty and the significance of this work.
> > Compared to Swin Transformer, the major contribution of this work should be a newly proposed window shifting mechanism for image restoration.
> > Basing on the comparison to SwinIR, it's hard to say the performance gains are brought by the proposed mechanism or the increases of parameters and Flops.
> > However I think this version deserves a borderline score.

---

> > > ### Author Response · Authors · 2022-08-09
> > > **Concerns have been acctually covered by the rebuttal response and main paper**
> > >
> > > `Diss-Q2-1:` *Sorry for the late. But I have to say that the openreivew system is very unfriendly for reviewers. I spent lots of time to find a way to access my comments and make a response.*
> > >
> > > `Diss-A2-1:` Thanks for spending time familiarizing the use of the openreview system and responding to our discussions.
> > >
> > >
> > >
> > > `Diss-Q2-2:` *Thanks for author's response. I see that authors provided multiple additional experiments to support their contribution.*
> > >
> > > `Diss-A2-2:` Thanks for reading the experiments we did in our rebuttal. And thanks for recognizing these work for supporting our contributions.
> > >
> > >
> > >
> > > `Diss-Q2-3:` *However, I still concern about the novelty and the significance of this work. Compared to Swin Transformer, the major contribution of this work should be a newly proposed window shifting mechanism for image restoration.*
> > >
> > > `Diss-A2-3:` **For "the major contribution"**. In addition to a newly proposed window shifting mechanism, there are other innovations and contributions. We have responded to some **similar** questions, like `Q1-2`, `Q2-1`, and `Q3-4`. You can refer to `A1-2`, `A2-1`, and `A3-4`. However, we still list more specifics here.
> > >
> > > A novelty self-attention mechanism: rectangle window self-attention (**Rwin-SA**) and locality complementary module (**LCM**).
> > >
> > > Compared to Swin Transformer, the above three designs are equally important for image restoration. We demonstrate the effect of three components in our **ablation study**.
> > >
> > > We also show them here (**Table 1(a) and Table 1(b) in the main paper**). Output size is 3$\times$256$\times$256 to calculate FLOPs.
> > >
> > > | Network                   |   PSNR    |    SSIM    | FLOPs  |
> > > | ------------------------- | :-------: | :--------: | :----: |
> > > | Square w/o shift          |   32.50   |   0.9325   | 281.8G |
> > > | Square w/ shift           |   32.75   |   0.9347   | 281.8G |
> > > | Rectangle w/o axial-shift | **32.66** | **0.9334** | 281.8G |
> > > | Rectangle w/ axial-shift  | **32.91** | **0.9360** | 281.8G |
> > >
> > > **(1)** Under **the same complexity**, Rwin-SA outperforms the square window attention (local window in Swin Transformer). Moreover, with axial-shift, our method obtains **0.16 dB** gain over the square window with shift operation in Swin Transformer.
> > >
> > > | Network       |   PSNR    |    SSIM    | FLOPs  |
> > > | ------------- | :-------: | :--------: | :----: |
> > > | CAT-R w/o LCM |   32.91   |   0.9360   | 281.8G |
> > > | CAT-R w/ LCM  | **32.98** | **0.9361** | 282.7G |
> > > | CAT-A w/o LCM |   33.01   |   0.9354   | 349.7G |
> > > | CAT-A w/o LCM | **33.11** | **0.9363** | 350.6G |
> > >
> > > **(2)** The LCM module achieves a performance improvement while slightly increasing the model complexity.
> > >
> > >
> > >
> > > `Diss-Q2-4:` *Basing on the comparison to SwinIR, it's hard to say the performance gains are brought by the proposed mechanism or the increases of parameters and Flops. However I think this version deserves a borderline score.*
> > >
> > > `Diss-A2-4:` **For "the comparison to SwinIR"**. We have responded to this question in `Q1-1`, `Q3-1`, and `Q4-2`. You can refer to `A1-1`, `A3-1`, and `A4-2`. However, we still list the specifics here.
> > >
> > > We provide a comparison of similar model sizes and FLOPs in **supplementary material** Table 2. And we have merged the results (**CAT-R-2**) into **Table 4** in the main paper. We also show them here. Output size is 3$\times$512$\times$512 to calculate FLOPs.
> > >
> > > | Method  | Params (M) | FLOPs (G) |   Set5    |   Set14   |   B100    | Urban100  | Manga109  |
> > > | ------- | :--------: | :-------: | :-------: | :-------: | :-------: | :-------: | :-------: |
> > > | SwinIR  |   11.90    |   215.3   | **32.92** |   29.09   |   27.92   |   27.45   |   32.03   |
> > > | CAT-R-2 |   11.93    |   216.3   |   32.91   | **29.13** | **27.93** | **27.59** | **32.16** |
> > >
> > > | Method         | EDSR  | RCAN  |  HAN  |  CSNLN   | SwinIR | CAT-R (ours) | CAT-A (ours) | CAT-R-2 (ours) |
> > > | -------------- | :---: | :---: | :---: | :------: | :----: | :----------: | :----------: | :------------: |
> > > | PSNR (dB)      | 26.64 | 26.82 | 26.85 |  27.22   | 27.45  |    27.62     |    27.89     |   **27.59**    |
> > > | FLOPs (G)      | 823.3 | 261.0 | 269.1 | 84,155.2 | 215,3  |    292.7     |    360.7     |     216.3      |
> > > | Parameters (M) | 43.09 | 15.59 | 16.07 |   6.57   | 11.90  |    16.60     |    16.60     |     11.93      |
> > >
> > > Our CAT-R-2 obtains a similar model size, FLOPs with SwinIR but obtains better performance. Therefore, the performance gains come from our proposed methods.
> > >
> > >
> > >
> > > **Please let us know if you have any unsolved or other concerns. Thanks.**

---

### Official Review · Reviewer_mWKh · 2022-07-11

**Rating:** 8
**Confidence:** 5
**Soundness:** 4 excellent
**Presentation:** 4 excellent
**Contribution:** 4 excellent

**Summary:**

A new Transformer based method, named as cross aggregation Transformer (CAT), was proposed for high-quality image restoration. The authors proposed three main components: rectangle-window self-attention, axial-shift operation, and locality complementary module (LCM). Their effects are demonstrated by extensive ablation study results. The main comparisons show the superior of the proposed CAT.

**Questions:**

- For JPEG artifact reduction, the performance improvements of CAT over SwinIR are not very obvious, although the visual differences are obvious. Please give some explanations.

- The authors claim that the method is for image restoration. Then, how would the CAT perform for other image restoration tasks? Will it still achieve SOTA performance?

- Do all the training and testing details between SwinIR and CAT for image SR and JPEG artifact reduction keep the same? If there has some differences, please clarify them.

- Please tell the differences between CAT and CSwin [10].

**Limitations:**

In supplementary file, the limitations and potential negative societal impact of the work have been briefly discussed.

**Strengths And Weaknesses:**

Strength

- The authors proposed to aggregate deep features across different windows with expanded sensing area, which is reasonable and effective for image restoration.

- A new shift operation named axial-shift operation was proposed to further increase the interaction of different windows.

- A Conv operation named locality complementary module (LCM) was proposed to make good use of its translation invariance and locality, enabling the model to capture both global and local information.

- Experiments are extensive. The ablation study shows the effect of each proposed new component. The main quantitative and visual comparisons with recent methods, like SwinIR, demonstrate that the proposed CAT is a promising Transformer based method.

- The authors provided source code and pretrained models for results reproduction. These materials could help other researchers for further investigations and improvements. For example, it would be easy to apply for this method for image denoising, derain, dehazing et al. with the provided code.

- The authors also show some visualization results using LAM and compare with SwinIR. According to the visualization results (Fig. 2 in supp.), the proposed CAT captures a larger receptive field. This observation is consistent with the claim in ‘Abstract’ about the long-range dependencies estimation.

- The main paper and supplementary file are well prepared. The writing is pretty good and easy to read. The motivation is clear and reasonable. The paper is carefully organized.

Weakness

- In Table 4, the parameter number and FLOPs of CAT are larger than those in SwinIR. A better and more reasonable way is to keep similar model size and FLOPs and then show comparisons.

- The detailed differences between axial-shift and the shift option in Swin Transformer are not very clear to me.

---

> ### Author Response · Authors · 2022-08-02
> **Response to Reviewer mWKh (denoted as R1) part 1**
>
> `Q1-1:` In Table 4, the parameter number and FLOPs of CAT are larger than those in SwinIR. A better and more reasonable way is to keep similar model size and FLOPs and then show comparisons.
>
> `A1-1:` Thanks for your suggestion. We **already provided** a comparison of similar model sizes and FLOPs in **supplementary material** Table 2. We also show them here. Output size is 3$\times$512$\times$512 to calculate FLOPs.
>
> | Method  | Params (M) | FLOPs (G) |   Set5    |   Set14   |   B100    | Urban100  | Manga109  |
> | ------- | :--------: | :-------: | :-------: | :-------: | :-------: | :-------: | :-------: |
> | SwinIR  |   11.90    |   215.3   | **32.92** |   29.09   |   27.92   |   27.45   |   32.03   |
> | CAT-R-2 |   11.93    |   216.3   |   32.91   | **29.13** | **27.93** | **27.59** | **32.16** |
>
> Our CAT-R-2 obtains a similar model size, FLOPs with SwinIR but obtains better performance.
>
> We have merged the results (CAT-R-2) into Table 4. We also show them here.
>
> | Method         | EDSR  | RCAN  |  HAN  |  CSNLN   | SwinIR | CAT-R (ours) | CAT-A (ours) | CAT-R-2 (ours) |
> | -------------- | :---: | :---: | :---: | :------: | :----: | :----------: | :----------: | :------------: |
> | PSNR (dB)      | 26.64 | 26.82 | 26.85 |  27.22   | 27.45  |    27.62     |    27.89     |     27.59      |
> | FLOPs (G)      | 823.3 | 261.0 | 269.1 | 84,155.2 | 215,3  |    292.7     |    360.7     |     216.3      |
> | Parameters (M) | 43.09 | 15.59 | 16.07 |   6.57   | 11.90  |    16.60     |    16.60     |     11.93      |
>
>
>
> `Q1-2:` The detailed differences between axial-shift and the shift option in Swin Transformer are not very clear to me.
>
> `A1-2:`  Thanks for asking for those details. We clarify them as follows.
>
> **(1)** Our axial-shift references the design of the shift option in Swin Transformer. However, our axial-shift adopts a grouped parallel design. Axial-shift is divided into V-Shift and H-Shift operations, which act on different attention heads and correspond to V-Rwin and H-Rwin. However, the shift operation in Swin Transformer performs the same shift operation in all heads.
>
> **(2)** Based on axial-shift, Rwin can achieve more window interactions, thereby expanding the receptive field and improving model performance. We can find that the performance of Rwin with axial-shift is much better than the square window with shift operation in Swin Transformer from **ablation study** Table 1 (a). We show them here. Output size is 3$\times$256$\times$256 to calculate FLOPs, and PSNR values are tested on Urban100 ($\times$2).
>
> | Network                   |   PSNR    |    SSIM    | FLOPs  |
> | ------------------------- | :-------: | :--------: | :----: |
> | Square. w/ shift          |   32.75   |   0.9347   | 281.8G |
> | Rectangle. w/ axial-shift | **32.91** | **0.9360** | 281.8G |
>
> **(3)** Furthermore, the shift operation in Swin Transformer can be viewed as a special case of our axial-shift. When the axial-shift displacement distances are the same in all heads, the shift operation in each attention head is the same. Then axial-shift degenerates into the shift operation in Swin Transformer. Overall, our axial-shift is more general and efficient.
>
>
>
> `Q1-3:` For JPEG artifact reduction, the performance improvements of CAT over SwinIR are not very obvious, although the visual differences are obvious. Please give some explanations.
>
> `A1-3:` Thanks for your question. We explain it below.
>
> **For "the visual differences are obvious".** The proposed CAT uses rectangle window self-attention, which can capture different features in horizontal and vertical directions for each pixel. This property enables CAT to recover more directional and repetitive texture features. So in the visual comparison of Fig. 6, CAT can recover more texture information, and the visual effect is better than SwinIR.
>
> **For "performance improvements of CAT over SwinIR are not very obvious".** **(1)** For the JPEG artifact reduction testing datasets: LIVE1 and Classic5, the number of images they contain is small (5 and 29), and the texture features are not rich. So the overall improvement effect is not obvious. Nevertheless, the difference is evident from the visual comparison (like Fig. 6 carnivaldolls).
>
> **(2)** In the **supplementary material** Table 3, we test CAT and SwinIR on the Urban100 dataset, which with more directional and repetitive texture features. The performance of CAT is much better than that of SwinIR, with a **0.25 dB** improvement. We also show them here (only PSNR).
>
> | Method |   q=10    |   q=20    |   q=30    |   q=40    |
> | ------ | :-------: | :-------: | :-------: | :-------: |
> | SwinIR |   30.55   |   33.12   |   34.58   |   35.50   |
> | CAT    | **30.80** | **33.38** | **34.81** | **35.73** |

---

> ### Author Response · Authors · 2022-08-02
> **Response to Reviewer mWKh (denoted as R1) part 2**
>
> `Q1-4:` The authors claim that the method is for image restoration. Then, how would the CAT perform for other image restoration tasks? Will it still achieve SOTA performance?
>
> `A1-4:` There are two ways to use CAT for other image restoration tasks:
>
> **(1)** Use the model architecture shown in Fig. 1. For different tasks with different reconstruction modules, keep the other modules unchanged. For example, for image denoising and deblurring, the reconstruction module is consistent with the one for JPEG artifact reduction.
>
> **(2)** Apply our proposed cross aggregation Transformer block to the encoder-decoder-based UNet architecture (like Restormer).
>
> We use the **second** way to build the CAT model and apply it to real image denoising. Our CAT outperforms or is comparable to other methods.
>
> We use SIDD training dataset to train our CAT. The training settings and training process are the same as Restormer. We test our model on SIDD and DND datasets. The comparison of our CAT with the state-of-the-art methods: MIRNet, Uformer, and Restormer is shown in the following table.
>
> | Method                                                       | Provenance | Params |   SIDD    |           |    DND    |           |
> | ------------------------------------------------------------ | ---------- | :----: | :-------: | :-------: | :-------: | :-------: |
> |                                                              |            |        |   PSNR    |   SSIM    |   PSNR    |   SSIM    |
> | MIRNet                                                       | ECCV 2020  | 31.79M |   39.72   |   0.959   |   39.88   |   0.956   |
> | Uformer-S [used for comparison in Table 6 of Restormer]      | CVPR 2022  | 20.63M |   39.77   |   0.959   |   39.96   |   0.956   |
> | Uformer-B [results shown in CVPR 2022 camera ready]          | CVPR 2022  | 50.88M |   39.89   |   0.960   |   40.04   |   0.956   |
> | Restormer [training iterations: 300K]                        | CVPR 2022  | 26.11M | **40.02** | **0.960** |   40.03   |   0.956   |
> | CAT [ours, **completed trained**; training iterations: 300K] | our method | 25.77M |   40.01   | **0.960** | **40.05** | **0.956** |
> | CAT+ [ours, **self ensemble**]                               | our method | 25.77M | **40.05** | **0.960** | **40.08** | **0.956** |
>
>
>
> We can observe that our CAT outperforms MIRNet, Uformer-S, and Uformer-B on all datasets. And our CAT performs better than Restormer on DND and is comparable to Restormer on SIDD. Moreover, the parameter of our CAT is smaller than all models compared. Overall, our CAT outperforms or is comparable to other methods, except Uformer-S.
>
> We have added this part to the supplementary material. Please refer to **Sec. 4.1** in the **supplementary material** for details. Moreover, we have updated the test code and pretrained model to our source code (in **Anonymous Github**) for reviewers to test and check.
>
> We also plan to apply our CAT to more image restoration tasks in the future.
>
>
>
> `Q1-5:` Do all the training and testing details between SwinIR and CAT for image SR and JPEG artifact reduction keep the same? If there has some differences, please clarify them.
>
> `A1-5:` Thanks for asking for those details. There is no difference in the training and testing details between CAT and SwinIR.

---

> ### Author Response · Authors · 2022-08-02
> **Response to Reviewer mWKh (denoted as R1) part 3**
>
> `Q1-6:` Please tell the differences between CAT and CSwin [10].
>
> `A1-6:` The main differences can be summarized into two points.
>
> **(1)** Axial-Rwin is inspired by CSwin and is close to CSwin in form, but it is only a special case of our proposed rectangle window self-attention. Regular-Rwin we proposed is a more general form and more flexible. In other words, the attention mechanism in CSwin is a special case proposed in our paper.
>
> **(2)** We also propose two operations: axial-shift and LCM module, to enhance Rwin-SA.
>
> We compare the differences between CAT (CAT-A, CAT-R) and CSwin in the following table. And replace our Cross Aggregation Transformer Block with CSWin Transformer Block proposed by CSwin in the CAT model to form a Transformer model. The implementation of CSwin is consistent with CAT-A. We train and test on image SR (×2). FLOPs are measured when the output size is set to 3×256×256, and PSNR values are tested on Urban100 (×2).
>
> | Method |            attention mechanism            | shift operation | LCM  | Parameters |   FLOPs   |   PSNR    |
> | ------ | :---------------------------------------: | :-------------: | :--: | :--------: | :-------: | :-------: |
> | CSwin  |    cross-shaped window self-sttention     |        x        |  x   |   16.45M   |   350.7   |   33.73   |
> | CAT-A  |  (axial) rectangle-window self-attention  |        √        |  √   |   16.46M   | **350.7** | **34.26** |
> | CAT-R  | (regular) rectangle-window self-attention |        √        |  √   |   16.46M   | **282.7** | **34.08** |
>
> We can see that the method we proposed is more suitable for image restoration tasks than CSwin.

---

> ### Author Response · Authors · 2022-08-05
> **Follow-up discussions with Reviewer R1 (mWKh)**
>
> Dear Reviewer mWKh ,
>
> Thanks for your valuable time and comments. We have responded to the related questions.
>
>
>
> **(1)** We provide **a variant CAT** of similar model size and FLOPs in supplementary material and merge the result into **Table 4** in the main paper.
>
> **(2)** We clarify the differences between **axial-shift and shift operation** in Swin Transformer.
>
> **(3)** We explain the **small improvement** for JPEG artifact reduction.
>
> **(4)** We apply our method to **real Image Denoising**. The performance of our CAT is **comparable to or better** than other methods (**MIRNet, Uformer, and Restormer**).
>
> **(5)** Finally, we provide analysis and experiments to show the differences between our CAT and **CSwin**.
>
>
>
> Please let us know if you have any unsolved or other concerns. Then, we have enough time to provide further feedback. Thanks.
>
> Best,
>
> Authors

---

> > ### Comment · Reviewer_mWKh · 2022-08-07
> > **Response to Authors' Rebuttal**
> >
> > Thanks for providing so-detailed responses including extensive experiments. My concerns have been well addressed as follows.
> >
> > (1) For fair comparisons with similar model size and FLOPs, I am satisfied with the provided results.
> >
> > (2) For other image restoration tasks, I appreciate that the authors provide real denoising results and compare with recent works, like Uformer and Restormer. The proposed CAT achieves good results.
> >
> > (3) The authors also compared with CSWin and discuss the differences with shift operation in Swin Transformer.
> >
> > I also read other reviewers comments and the corresponding responses carefully. The authors provide training log and models in supplementary file, which makes the work more solid. So, I would like to upgrade my rating and vote for acceptance.

---

### Author Response · Authors · 2022-08-02
**Response to all reviewers**

We thank all reviewers for their valuable time and comments. Based on the comments of four reviewers, we have modified the main paper and added additional experiments and analyses in the supplementary material.

**For the main paper.**

1. We add the inspiration of CSwin for axial-Rwin in introduciton and method.
2. We discuss more Transformers in related work, including CSwin and Uformer.
3. We add a more reasonable comparison in model size analyses.

**Note:** In the latest submitted paper, the modifications are marked in blue font.



**For the supplementary material.**

1. Experiment on real image denoising.
2. Experiment on color  JPEG compression artifact reduction.
3. Other numerical results, including the comparison between CSwin and CAT.
4. Analysis of difference between axial-shift and shift operation in Swin Transformer.

---

### Author Response · Authors · 2022-08-09
**Response to all reviewers and area chairs**

We thank all reviewers and area chairs for their valuable time and comments. After discussing with reviewers and providing more clarifications/results/analyses, we would like to give a brief response.

Reviewer mWKh (denoted as R1), Reviewer LKyp (denoted as R3), and Reviewer J9TR (denoted as R4) all hold a **positive** side for our work. Our responses have covered their questions.

Although Reviewer RMwD (denoted as R2) gives a negative score in the first round, R2 upgrades his rating and approves our new experiments after the rebuttal. For further concerns of R2, we find that all those concerns have been well addressed in our rebuttal and the main paper. R2 can easily find the corresponding responses.

The source code and related pretrained models have been provided in supplementary file and anonymous site. We will also make all of them public soon!

We thank all reviewers and area chairs again!

Best,

Authors

---

### Meta-Review · Area_Chair_QxN3 · 2022-08-26

**Recommendation:** Accept
**Confidence:** Less certain

**Metareview:**

This paper proposes a cross aggregation transformer for image restoration. The Rwin-SA with axial-shift is introduced to aggregates the features cross different windows and the locality complementary module (LCM) is introduced to capture both local and global information. Massive experiments on different datasets and tasks demonstrate that the CAT outperforms the state-of-art methods. Several concerns are raised by the reviewers including the novelty, discussion and experiments. After an in-depth discussion between the reviewers and authors, three reviewers hold a positive side (strong accept) for this work, and reviewer RMwD increases the rating to a borderline reject and approves the new experiments provided in the rebuttal. Considering the average score and the contribution of this work, the AC recommends acceptance. The AC strongly urges the authors to consider all the comments in preparing the final version.

**Award:**

No

---

### Decision · Program_Chairs · 2022-09-14

Accept